# MODEL RISK-SENSITIVE OFFLINE REINFORCEMENT LEARNING

**Gwangpyo Yoo, Honguk Woo**[*]
Department of Computer Science and Engineering, Sungkyunkwan University
{necrocathy, hwoo}@skku.edu

## ABSTRACT

Offline reinforcement learning (RL) is becoming critical in risk-sensitive areas such as finance and autonomous driving, where incorrect decisions can lead to substantial financial loss or compromised safety. However, traditional risk-sensitive offline RL methods often struggle with accurately assessing risk, with minor errors in the estimated return potentially causing significant inaccuracies of risk estimation. These challenges are intensified by distribution shifts inherent in offline RL. To mitigate these issues, we propose a model risk-sensitive offline RL framework designed to minimize the worst-case of risks across a set of plausible alternative scenarios rather than solely focusing on minimizing estimated risk. We present a critic-ensemble criterion method that identifies the plausible alternative scenarios without introducing additional hyperparameters. We also incorporate the learned Fourier feature framework and the IQN framework to address spectral bias in neural networks, which can otherwise lead to severe errors in calculating model risk. Our experiments in finance and self-driving scenarios demonstrate that the proposed framework significantly reduces risk, by $11.2\%$ to $18.5\%$, compared to the most outperforming risk-sensitive offline RL baseline, particularly in highly uncertain environments.

## 1 INTRODUCTION

Many application domains of offline reinforcement learning (RL), such as finance (Zhang et al., 2020; Wang & Ku, 2022), self-driving (Bernhard et al., 2019; Seres et al., 2023), and healthcare (Lu et al., 2020), are inherently risk-sensitive due to the potential for high costs or safety risks from interactions between the agent and the environment at deployment. In these scenarios, both premature and well-trained agents can make unsafe decisions, prompting active research efforts to integrate offline RL with risk-sensitive RL (Urpí et al., 2021; Ma et al., 2021; Rigter et al., 2024). These efforts aim to minimize risk instead of maximizing expected return in offline RL settings. Here, *(aleatoric) risk*[1] is defined as a real number representing the severity of potential harm associated with the return distribution. The function that maps a return distribution to its corresponding risk is called *risk measure*. For example, the risk calculated by conditional value at risk (*CV@R*) risk measure with a confidence level $\alpha \in [0, 1]$ is the negation of the average from the most pessimistic to the $\alpha$-th quantile (Yoo et al., 2024).

As these risk measures are designed to emphasize rare events, while underestimating frequent ones, the estimated risk becomes highly sensitive to changes in the underlying return distribution (Cont et al., 2010; Kou et al., 2013; Embrechts et al., 2015; Pesenti et al., 2016). Such errors can lead to incorrect risk management, especially when there is a distribution shift between the training and deployment environments. This situation is common in offline RL, where the agent cannot receive real-time feedback from the environment.

In finance, the issue is addressed by minimizing the worst-case of risks across a set of plausible alternative scenarios rather than directly minimizing the risk estimated by the model (Bernard et al., 2023). This worst-case is referred to as *model risk*, which quantifies the worst-case consequence of

---

[*]Corresponding Author
[1]Whenever the context is clear, we will refer to aleatoric risk as risk.

Figure 1: Difference between risk and model risk measured by $CV@R_\alpha$. The negation of the shaded area represents the risk.

using incorrect models (Breuer & Csiszár, 2016; Bernard et al., 2023). Unlike traditional approaches that rely solely on the estimated risk and are vulnerable to model errors, the model risk acknowledges the potential for inaccuracies in the model, enabling more robust and reliable risk-sensitive decision-making. Inspired by this, our work in risk-sensitive offline RL shifts the focus from conventional risk minimization to model risk minimization. In doing so, we address the common issue of estimating return distribution statistics (mean and the standard deviation) in RL contexts, where the statistics are often unreliable or unknown. This contrasts with finance, where the mean and deviation are assumed to be known.

Figure 1 illustrates the difference between traditional risk-sensitive offline RL approaches (left-side in the figure) and our model risk-sensitive offline RL (right-side in the figure). In the left-side, existing risk-sensitive offline RL approaches minimize the risk estimated by critic (green area). In the right-side, our model risk-sensitive RL minimizes risk (blue area) of the worst-case scenario (blue line) among the plausible alternative scenarios set (gray lines). By minimizing the model risk, the agent accounts for the possibility of being incorrect, aiming to achieve the best possible results despite potential errors.

We present a model risk-sensitive offline RL framework that accounts for model errors, enhancing robustness in highly uncertain environments. To the best of our knowledge, this is the first approach that incorporates model risk into offline RL algorithms. Specifically, we develop a critic-ensemble criterion method that identifies plausible alternative scenarios, enabling the calculation of model risk without introducing additional hyperparameters. Achieving this requires accurate return distribution statistics, particularly the mean and deviation. To enhance their precision, which is essential for identifying plausible alternative scenarios, we adopt a Fourier feature network for quantile regression. Through case studies in finance and self-driving scenarios, we demonstrate that our model risk-sensitive offline RL framework is more robust than other approaches, achieving an 11.2% to 18.5% reduction in risk. Notably, our approach exhibits strong advantages in highly uncertain environments with a large possibility of model errors.

Our contributions are as follows.

- We present the model risk-sensitive offline framework that minimizes the model risk, enabling more robust decision-making compared to conventional risk-sensitive offline RL approaches.
- We introduce the critic-ensemble criterion, which adjusts a set of alternative scenarios by expanding or contracting it based on the level of uncertainty, thereby enabling the computation of model risk.
- We devise the Fourier feature quantile regression network to precisely estimate the statistics of distributions, which are required to calculate the model risk.
- Through several case studies, we demonstrate the robustness of our approach in highly uncertain environments.

## 2 RELATED WORK

**Risk-sensitive RL.** Traditional research on risk-sensitive RL is based on risk measures that can be computed without estimating whole return distributions (Howard & Matheson, 1972; Sato et al.,

2001; Mihatsch & Neuneier, 2002; Tamar et al., 2015; Chow et al., 2017). With the rise of distributional RL, Dabney et al. (2018a) developed IQN framework, which enables the direct computation of risk via quantile regression, thereby unifying the risk-sensitive RL framework. Other approaches include the direct calculation of *CV@R* policy gradient without calculating the return distribution (Tamar et al., 2015; Chow et al., 2017), entropy-based methods (Mihatsch & Neuneier, 2002), dynamic *CV@R*(Du et al., 2022; Lim & Malik, 2022), and entropic value at risk (*EV@R*, Ni & Lai (2022)). Dynamic *CV@R* recursively accounts for risk at each timestep and is related to 1R2R (Rigter et al., 2024). *EV@R* is a risk measure that is out of our scope, but it is closely related to CODAC (Ma et al., 2021). Another interesting approach is integrating risk-sensitive RL with safe RL. For example, Kim & Oh (2022) integrate the risk-sensitive approach to the safe RL, constraining the *CV@R* of costs instead of expected costs. Meanwhile, Ying et al. (2021); Greenberg et al. (2022) constrain the risk using the safe RL method, while maximizing return. In an online RL setting, Jaimungal et al. (2022) have proposed a similar approach to ours. However, because their method is based on KDE and on-policy algorithm, therefore, it cannot be applied to offline RL.

**Risk-sensitive Offline RL.** Research in risk-sensitive offline RL relies on integrating IQN and offline RL frameworks. Urpí et al. (2021) first proposed a risk-sensitive offline RL framework ORAAC by combining IQN and BCQ (Fujimoto et al., 2019). ORAAC is hindered by excessive constraints on the behavior policy, limiting its capability to generate better trajectories. To address the limitation, CODAC (Ma et al., 2021), which is based on CQL, was proposed. Yet, CODAC inherits the issues from CQL, such as hyperparameter sensitivity (Tarasov et al., 2024) [2]. Another approach, 1R2R (Rigter et al., 2024), is a model-based RL method that estimates risk with worst-case perturbations of the transition. However, its main limitation is that it is only guaranteed when transition probabilities follow a normal distribution.

**Model Risk.** Our work is built on the concept of model risk, which has been extensively studied in the field of mathematical finance (Bernard et al., 2023; Breuer & Csiszár, 2016; Blanchet & Murthy, 2019; Glasserman & Xu, 2014). Focusing on the worst-case expectation among scenarios aligns with Breuer's approach to aleatoric risk using a coherent risk measure (Breuer & Csiszár, 2016; Glasserman & Xu, 2014). However, we focus on the worst-case of *aleatoric risk* among alternative scenarios, following Bernard et al. (2023), which leads to a different formulation. This approach allows comparison of value distributions with different support, which is crucial in offline RL, since the worst-case distribution may not match the model's support (Bernard et al., 2023).

## 3 BACKGROUND AND PROBLEM FORMULATION

### 3.1 BACKGROUND

**Risk-sensitive RL.** We consider a Markov decision process $(\mathcal{S}, \mathcal{A}, R, \mathcal{P}, \gamma)$ where $\mathcal{S}$ is a set of states, $\mathcal{A}$ is a set of actions, $R : \mathcal{S} \times \mathcal{A} \to \triangle(\mathbb{R})$ is an immediate reward which may be random, and $\mathcal{P} : \mathcal{S} \times \mathcal{A} \to \triangle(\mathcal{S})$ is a transition probability. Here $\triangle(\mathcal{X})$ denotes a set of random variables whose supports are subsets of $\mathcal{X}$. Further, we assume that all random attributes' means and variances are finite. An optimal risk-sensitive policy is formulated as

$$\pi^* = \arg\min_{\pi} H_\phi\left(Z^\pi(s,a)\right) \text{ where } Z^\pi(s,a) = \sum_{t=0}^{T} R(s_t, a_t)\gamma^t | s_0 = s, a_0 = a. \quad (1)$$

Here $H_\phi(Z^\pi(s,a))$ denotes the risk of policy $\pi$ with respect to its return $Z^\pi(s,a)$ and $H_\phi$ denotes a risk measure given by the user or environment. As we are interested in spectral risk measures (Adam et al. (2008), in Def. 2), we use the quantile function of a random variable.

**Definition 1** *(Koenker, 2005) Let $Z$ be a random variable. A quantile function of the random variable $Z$, $F_Z^{-1} : [0, 1] \to \mathbb{R}$, is defined as the left inverse of cumulative distribution of $Z$. Specifically,*

$$F_Z^{-1}(p) := \inf\{x \in \mathbb{R} \mid p \leq F_Z(x)\}, \quad (2)$$

*where $F_Z : \mathbb{R} \to [0, 1]$ is the cumulative distribution function of $Z$. For readability, we denote $F_Z^{-1}(\cdot; s, a)$ as the quantile function of dependent random variable, $Z|s,a$, rather than $F_{Z|s,a}^{-1}(\cdot)$.*

---

[2]This is also a problem commonly seen in GAN-based approaches. See Appendix for an explanation of the relationship between CQL and GAN.

The spectral risk is the weighted integral of the quantile function.

**Definition 2** *(Adam et al., 2008) A function $H_\phi : L^1 \to (\mathbb{R} \cup \{\infty\})$ is called spectral risk measure which has the following representation*

$$H_\phi(Z) = -\underbrace{\int_0^1 F_Z^{-1}(p)\phi(p)dp = -\int_0^1 F_Z^{-1}(F_\phi^{-1}(u))du}_{u := F_\phi(p) \Rightarrow du = dF_\phi(p) = \phi(p)dp \text{ with } p = F_\phi^{-1}(u)} = -\mathbb{E}_{u \sim U[0,1]}[F_Z^{-1}(F_\phi^{-1}(u))], \quad (3)$$

*where $\phi : [0, 1] \to [0, \infty]$ is a weight function; right-continuous, non-increasing with $\int_0^1 \phi(p)dp = 1$; i.e, $\phi$ is a density function for some random variable. Further, the value of this function given $Z$, $H_\phi(Z)$, is called the risk of $Z$. Here $L^1$ is a set of random variables whose means are finite. The right most term is followed by and inverse transform sampling (Miller et al., 2010; Dabney et al., 2018a) and Monte-Carlo integral.*

Since $\phi$ is non-increasing, the risk measure necessarily prioritizes worse outcomes, as intended. The negation is required in the definition, since we aim to *minimize* the risk. For example, when we want to optimize for the worst-case risk measure (i.e., $\phi$ is a Dirac-delta function), we minimize the negation of the worst-case outcome, the risk.

To calculate the risk $H_\phi(Z^\pi(s,a))$, we use the quantile function of $Z^\pi(s,a)$, $F_{Z^\pi}^{-1}(\cdot, s, a)$. To learn the quantile function, we use the following distributional Bellman equation (Bellemare et al., 2017; Dabney et al., 2018b;a).

$$Z^\pi(s_t, a_t) \stackrel{\text{distr.}}{=} R(s_t, a_t) + \gamma Z^\pi(s_{t+1}, a), \quad (4)$$

where $a = \arg\min_{a'} H_\phi(Z(s_{t+1}, a'))$. Specifically, we use the quantile regression method to estimate the quantile of return $Z^\pi(s,a)$. The quantile regression loss (Koenker, 2005; Dabney et al., 2018b) of the critic is

$$\mathcal{L}_{\text{crit.}}(\theta) = \frac{1}{NN'} \sum_{j=1}^{N'} \sum_{i=1}^{N} \underbrace{|p_i - \mathbf{1}(\delta_{ij}^\theta < 0)|}_{\text{Asymmetric weighted}} \cdot \underbrace{|\delta_{ij}^\theta|}_{L^1 \text{loss}}, \quad (5)$$

where $\delta_{ij}^\theta$ is the distributional Bellman residual and $\theta$ is a critic's parameter. The distributional Bellman residual is defined as a pairwise difference between the target quantile and the critic's quantile as

$$\delta_{ij}^\theta = R(s_t, a_t) + \gamma F_{Z^\pi}^{-1}(p_j; s_{t+1}, a, \theta_{\text{tar}}) - F_{Z^\pi}^{-1}(p_i; s_t, a_t, \theta), \quad (6)$$

where $p_i$, $p_j$ denote $N, N'$ numbers of independent random variables which follow $U[0, 1]$ and $\theta_{\text{tar}}$ denotes the parameter of target-critic.

**Risk-sensitive Offline RL.** In offline settings, an agent is not allowed to interact directly with the environment and can only access a transition dataset $\mathcal{D} := \{(s_i, a_i, r_i, s_i')\}_{i=1}^{|\mathcal{D}|}$. Due to this constraint, naive RL approaches suffer from overestimation of $Z^\pi(s,a)$ from distribution shifts (Fujimoto et al., 2019). Thus, an auxiliary regulation is commonly applied. The objective of offline RL is defined as

$$\pi^* = \arg\max_\pi \mathbb{E}[Z^\pi(s,a)] \text{ constraint to } D(\pi \| \pi_\mathcal{D}) < \epsilon, \quad (7)$$

where $D(\cdot \| \cdot)$ is some divergence (or metric), $\pi_\mathcal{D}$ is the behavior policy for the dataset $\mathcal{D}$, and $\epsilon > 0$ is a tolerance. Incorporating this regulation, the objective of risk-sensitive offline RL is extended from Eq.(1) as

$$\pi^* = \arg\min_\pi H_\phi(Z^\pi(s,a)) \text{ constraint to } D(\pi \| \pi_\mathcal{D}) < \epsilon. \quad (8)$$

### 3.2 PROBLEM FORMULATION

The primary concern of risk-sensitive offline RL is that the error of computing $H_\phi(Z^\pi(s,a))$ can be significant, as the agent cannot receive direct feedback from the environment in offline settings. To address this, we instead compute the worst-case of risks among alternative scenarios (i.e., model risk) and focus on minimizing the model risk. The model risk is defined as the highest possible risk within a $\sqrt{\varepsilon}$-ball with mean $\mu$ and standard deviation $\sigma$. The random variables in the ball specify the plausible alternative scenarios.

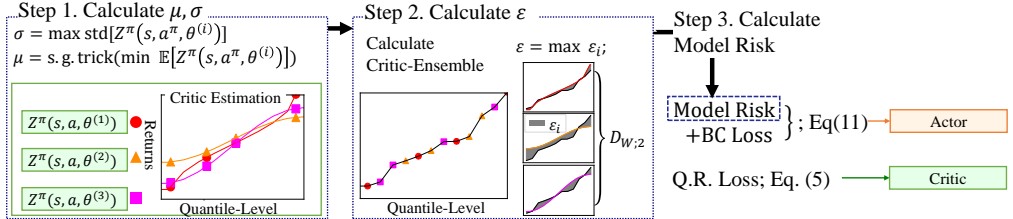

Figure 2: Overall framework of MR-IQN. Q.R. loss is abbreviation of quantile regression loss and s.g. trick is stop-gradient trick. Note that the stop-gradient trick does not change the value.

**Definition 3** *(Bernard et al. (2023)) The model risk $MR_\phi(Z^\pi(s,a);\mu,\sigma,\varepsilon)$ is defined as*

$$MR_\phi(Z^\pi(s,a);\mu,\sigma,\varepsilon) := \sup_{X \in M(Z^\pi(s,a);\mu,\sigma,\varepsilon)} H_\phi(X), \tag{9}$$

*where $M(Z^\pi(s,a);\mu,\sigma,\varepsilon) = \{X \in L^2 \mid \mathbb{E}[X] = \mu, \mathbb{V}ar[X] = \sigma^2, D_{W;2}(X, Z^\pi(s,a)) < \sqrt{\varepsilon}\}$ and $L^2$ is a set of random variables whose mean and variance are finite. Here, $D_{W;2}$ is the Wasserstein 2-distance defined as $D_{W;2}(X,Y) = \sqrt{\int_0^1 |F_X^{-1}(p) - F_Y^{-1}(p)|^2 dp}$. The distribution of random variable $Z^\pi(s,a)$ is called reference distribution.*

Our goal is to find a policy $\pi^*$ such that

$$\pi^* = \arg\min_\pi MR_\phi(Z^\pi(s,a);\mu,\sigma,\varepsilon) \text{ constraint to } D(\pi\|\pi_\mathcal{D}) < \epsilon. \tag{10}$$

## 4 MODEL RISK-SENSITIVE OFFLINE RL ALGORITHM

To achieve a model risk-sensitive offline RL agent satisfying the objective in Eq.(10), we introduce Model Risk IQN (MR-IQN), which integrates the TD3+BC framework (Fujimoto & Gu, 2021) with the model risk objective from Eq.(10). The MR-IQN consists of distributional critics and a deterministic actor. The critic is trained to minimize the loss function in Eq.(5), similar to other approaches (Urpí et al., 2021; Dabney et al., 2018a; Kuznetsov et al., 2020). Specifically, the critics are implemented using IQN (Dabney et al., 2018a) and TQC (Kuznetsov et al., 2020), following Yoo et al. (2024). Meanwhile the actor is trained to minimize the loss function,

$$\mathcal{L}(\pi) = \underbrace{\lambda_{\text{q.learn}} MR_\phi(Z^\pi(s,a);\mu,\sigma,\varepsilon)}_{\text{Model risk}} + \underbrace{(a - a_\mathcal{D})^2}_{\text{BC loss}}. \tag{11}$$

Here $\lambda_{\text{q.learn}} > 0$ is a scale parameter, $a_\mathcal{D}$ is a batch action, and $a = \pi(s)$.

Figure 2 describes the MR-IQN framework. We extract $\mu$ and $\sigma$ from the critic estimation. To precisely estimate the statistics, $\mu$ and $\sigma$, we leverage the Fourier feature architecture (Tancik et al., 2020) for the critics, as in Li & Pathak (2021). Next, we calculate $\varepsilon$ as the distance between the ensemble of the critics obtained by quantile mixture of individual critics. Finally, we calculate $MR_\phi(Z^\pi(s,\pi(s)))$ and minimize the loss function in Eq.(11). To do so, we first introduce the theorem and corollary below for calculating model risk.

**Theorem 1** *(Bernard et al., 2023) Let $Z$ be a random variable. For $\varepsilon > (\mu_Z - \mu)^2 + (\sigma_Z - \sigma)^2$, there exists a unique quantile function $h_\lambda : [0,1] \to \mathbb{R}$ which satisfies*

$$MR_\phi(Z;\mu,\sigma,\varepsilon) = -\int_0^1 h_\lambda(p)\phi(p)dp, \tag{12}$$

*where $\mu_Z$ is the mean of $Z$, $\sigma_Z$ is the standard deviation of $Z$. We have*

$$h_\lambda(p) = \mu - \sigma\left(\frac{\phi(p) + \lambda F_Z^{-1}(p) - a_\lambda}{b_\lambda}\right), \tag{13}$$

*where $\lambda \geq 0$ is some constant, and $a_\lambda = \mathbb{E}[\phi(U) + \lambda F_Z^{-1}(U)], b_\lambda = std[\phi(U) + \lambda F_Z^{-1}(U)]$. Here $U$ denotes the uniform random variable $U[0, 1]$ and std denotes the standard deviation operator. If $\varepsilon \leq (\mu_Z - \mu)^2 + (\sigma_Z - \sigma)^2$ holds, the problem is ill-posed or it has only a trivial solution.*

*Let $c_0 = corr(F_Z^{-1}(U), \phi(U))$. If $\varepsilon < (\mu_Z - \mu)^2 + (\sigma_Z - \sigma)^2 + 2\sigma\sigma_Z(1 - c_0)$ holds, we achieve*

$$\lambda = \frac{K}{\sigma_Z^2}\sqrt{\frac{V\sigma_Z^2 - C_{\phi,Z}^2}{\sigma^2\sigma_Z^2 - K^2}} - \frac{C_{\phi,Z}^2}{\sigma_Z}, \tag{14}$$

*where $V = \mathbb{V}ar[\phi(U)]$, $C_{\phi,Z} = \mathbb{C}ov[F_Z^{-1}(U), \phi(U)]$, and $K = \frac{1}{2}\left((\mu_Z - \mu)^2 + \sigma^2 + \sigma_Z^2 - \varepsilon\right)$. Otherwise, $\lambda = 0$. Here, corr is the Pearson correlation.*

**Corollary 1** *(Bernard et al., 2023) The solution of Eq.(9) exists and unique whenever $h_\lambda$ exists as*

$$MR_\phi(Z^\pi(s,a); \mu, \sigma, \varepsilon) = -(\underbrace{\mu - \sigma std[\phi(U)] corr[\phi(U), \phi(U) + \lambda F_{Z^\pi}^{-1}(U; s, a)]}_{\int_0^1 h_\lambda(p)\phi(p)dp \text{ in } Eq.(12)}), \tag{15}$$

*where $\lambda \geq 0$ is the constant determined by Eq.(14) in Theorem 1.*

## 4.1 CRITIC-ENSEMBLE CRITERION

To calculate the model risk for policy gradient using Eq.(15), it is required to determine $\mu, \sigma$ and tolerance $\varepsilon$. For $\varepsilon$, we want to set $\varepsilon$ large when there is a large inconsistency between an individual critic while setting $\varepsilon$ small when the inconsistency is small. For conservatism, we choose $\mu$ as the minimum value among the critics' estimation and $\sigma$ as the maximum to reduce overestimation bias as discussed by Fujimoto et al. (2018). From now on, we use $K$ to denote the number of critics.

**Step 1. Calculating $\mu, \sigma$.** To ensure conservatism, we select $\mu_0, \sigma$ as the smallest expectation and the largest deviation of $Z^\pi$, as follows.

$$\mu_0 := \min_{i=1,\dots,K} \mathbb{E}[Z^\pi(s, a, \theta^{(i)})], \sigma := \max_{i=1,\dots,K} std[Z^\pi(s, a, \theta^{(i)})] \tag{16}$$

When $\lambda = 0$, the gradient of the policy in Eq.(15) solely depends on the mean-deviation risk measure, which often leads to local optima. To mitigate this, we apply the stop gradient trick to $\mu$ as

$$\mu := -H_\phi(Z^\pi(s, a)) + \text{stop-grad}(H_\phi(Z^\pi(s, a))) + \mu_0, \tag{17}$$

where $H_\phi(Z^\pi(s, a)) = \max_{i=1,\dots,K} H_\phi(Z^\pi(s, a, \theta^{(i)}))$, i.e., the most conservative critic.

**Step 2. Calculating $\varepsilon$ through quantile mixture.** We construct an ensemble of critics as a single quantile function by mixing their quantiles and calculate the distance between this ensemble and individual critics.

We first form the critic ensemble by sorting the estimates from all critics and interpolating them over an equally divided space (i.e., quantile mixture). Next, we calculate the Wasserstein-2 distance between the ensemble of critics and individual critics. The largest of these distances is selected as $\varepsilon$.

The first step involves calculating the sorted critic values $y_k$ defined as

$$\{y_1, \dots, y_{NK}\} = \cup_{i=1}^K \cup_{j=1}^N F_{Z^\pi}^{-1}(p_j; s, a, \theta^{(i)}), \ y_1 \leq y_2 \leq \dots \leq y_{NK}, \tag{18}$$

where $N$ represents the number of random variables $p_j \sim U[0, 1]$. Let $X$ be an equally divided set of $[0, 1]$ defined as $X = \{x_1 = 0, x_2 = 1/(NK - 1), \dots, x_{NK} = 1\}$. By interpolating pairs $\{(x_1, y_1), \dots, (x_{NK}, y_{NK})\}$, we construct the ensemble of critics, $F_{\text{ens.}}^{-1}$. Figure 3 depicts the procedure of calculating $F_{\text{ens.}}^{-1}$. Finally, $\varepsilon$ is computed by

$$\varepsilon = \max_{i=1,\dots,K} \int_0^1 |F_{\text{ens.}}^{-1}(x; s, a) - F_{Z^\pi}^{-1}(x; s, a, \theta^{(i)})|^2 dx = D_{W;2}^2(F_{Z^\pi}^{-1}(\cdot; s, a, \theta^i), F_{\text{ens.}}^{-1}(\cdot)). \tag{19}$$

We use the trapezoid method to numerically calculate the integral in Eq.(19).

**Step 3. Calculate the model risk.** We take the most conservative critic and take the reference distribution of the critic estimates in the place of $F_Z^{-1}$ in Eq.(14) and Eq.(15).

$$F_{\text{ref}}^{-1}(\cdot) = F_{Z^\pi}^{-1}(\cdot; s, a, \theta^{(*)}), \text{ where } \theta^{(*)} = \arg\max_{\theta^{(i)}} H_\phi(Z^\pi(s, a, \theta^{(i)})). \tag{20}$$

---

**Algorithm 1** Critic-Ensemble Model Risk

---

**Inputs**: Critic Parameters $\theta^{(i)}$, Risk Measure's Density $\phi$, $N$ the number of samples.
**Output**: Model Risk $\text{MR}_\phi(Z^\pi(s,a),\mu,\sigma,\varepsilon)$
**Step1 Calculate $\mu,\sigma$:**
Sample $p_j$ from $U[0,1]$ with $j=1,\ldots,N$. ▷ To estimate risk and mean by inv. trans. sampling.
$\mu_0 \leftarrow \min_{i=1\ldots K} \mathbb{E}_{p_j}[F_{Z^\pi}^{-1}(p_j;s,a,\theta^{(i)})]$      ▷ Get minimal mean over critics; mean over $p_j$.
$p_j' \leftarrow F_\phi^{-1}(p_j), H_\phi(Z^\pi(s,a)) \leftarrow \max_{i=1,\ldots,K} -\mathbb{E}_{p_j'}[F_{Z^\pi}^{-1}(p_j';s,a,\theta^{(i)})])$      ▷ Eq.(3)
$\mu \leftarrow -H_\phi(Z^\pi(s,a)) + \text{stop-grad}(H_\phi(Z^\pi(s,a))) + \mu_0$    ▷ Stop-Gradient Trick in Eq.(17)
$\sigma \leftarrow \max_{i=1,\ldots,K} \text{std}_{p_j}[F_{Z^\pi}^{-1}(p_j,s,a,\theta^{(i)})]$      ▷ Get maximal standard deviation over critics.
**Step 2 Calculate $\varepsilon$:**
$\{y_k\}_{k=1}^{NK} \leftarrow \text{sort}(\cup_{i=1}^K \cup_{j=1}^N \{F_{Z^\pi}^{-1}(p_j;s,a,\theta^{(i)})\})$      ▷ Sort
$\{x_1 \leftarrow 0, x_2 \leftarrow \frac{1}{NK-1},\ldots,x_{NK-1} \leftarrow \frac{NK-2}{NK-1}, x_{NK} \leftarrow 1\}$      ▷ $x \leftarrow \text{linspace}(0,1,NK)$
$F_{\text{ens}}^{-1} \leftarrow \text{interpolation}(\{(x_k,y_k)\}_{k=1}^{NK})$      ▷ Rearrange and Interpolate
$\varepsilon = \max_{i=1,\ldots,K} \int_0^1 |F_{\text{ens}}^{-1}(x) - F_{Z^\pi}^{-1}(x;s,a,\theta^{(i)})|^2 dx$      ▷ Calculate via numerical integral.
**Step 3 Calculate Model Risk:**
$\theta^{(*)} \leftarrow \arg\max_{\theta^{(i)}} H_\phi(Z^\pi(s,a,\theta^{(i)})), F_{\text{ref}}^{-1} \leftarrow F_{Z^\pi}^{-1}(\cdot;s,a,\theta^{(*)})$   ▷ Take the most conservative.
$\lambda \leftarrow \text{calculate lambda}(F_{\text{ref}}^{-1},\mu,\sigma,\varepsilon,\phi)$      ▷ using Eq.(14).
$\text{MR}_\phi(Z^\pi(s,a)) \leftarrow \text{calculate model risk}(F_{\text{ref}}^{-1},\mu,\sigma,\varepsilon,\phi)$      ▷ using Eq.(15)
**Return:** $\text{MR}_\phi(Z^\pi(s,a))$

---

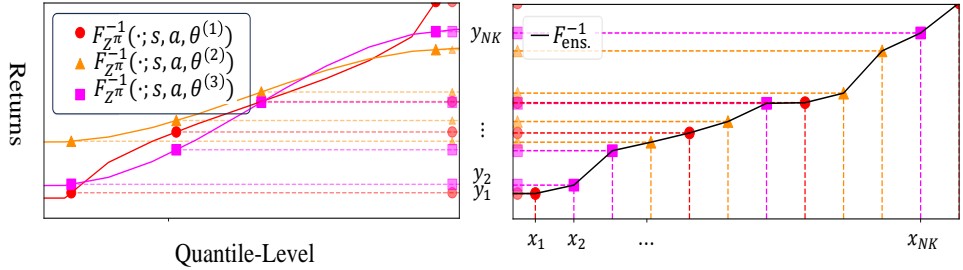

Figure 3: Procedure of calculating $F_{\text{ens}}^{-1}$ with three critics. $F_{\text{ens}}^{-1}$ is a quantile mixture of critics, therefore, it shows a mid-level pessimism similar to $F_{Z^\pi}^{-1}(\cdot,s,a,\theta^{(3)})$.

If a solution does not exist, (i.e., $\varepsilon < (\mu_Z - \mu)^2 + (\sigma_Z - \sigma)^2$), we substitute $H_\phi(Z^\pi(s,a))$ for $\text{MR}_\phi(Z^\pi(s,a);\mu,\sigma,\epsilon)$ to preserve the gradient signal.

Algorithm 1 outlines the procedure of computing model risk throughout the critic-ensemble criterion. First, the most conservative $\mu$, $\sigma$ among critics, are chosen. Next, we compute $\varepsilon$ by measuring the $D_{W;2}$ distance between each individual critic and the ensemble, which is obtained by quantile mixture via sort and rearrange. Finally, we compute the model risk using the parameters obtained from the previous steps.

## 4.2 ADDRESSING SPECTRAL BIAS OF QUANTILE REGRESSION

Although our framework accounts for model errors, it is essential to estimate the mean ($\mu$) and deviation ($\sigma$) as accurately as possible, since these statistics are assumed to approximate the ground truth. While quantile regression yields an unbiased estimator (Koenker, 2005), due to spectral bias (Rahaman et al., 2019), conventional neural networks often fail to capture these statistics effectively. Spectral bias refers to the phenomenon where neural networks struggle to learn high-frequency components of patterns. For risk-sensitive RL to be effective, $Z^\pi$ must converge in the distributional sense to ensure proper signals can be back-propagated. According to the Lévy convergence theorem (Williams, 1991), this is equivalent to the pointwise convergence of the Fourier transform of $Z^\pi$, emphasizing the critical role of the high-frequency domain (and thus spectral bias) in convergence. The Fourier feature network (Tancik et al., 2020; Li & Pathak, 2021) is a known solution to the spectral bias problem.

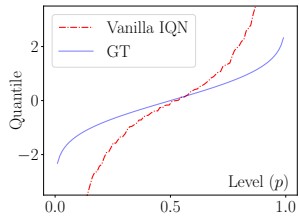 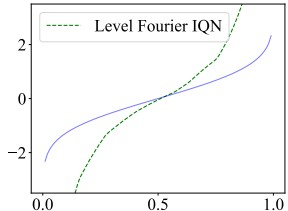 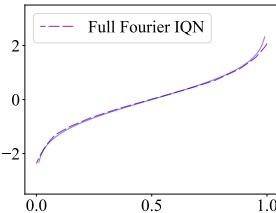

(a) Vanilla IQN
$D_{W;1}$: $2.23 \pm 0.05$
Crossing(%): $6.07 \pm 2.08$

(b) Fourier feature in level domain
$D_{W;1}$: $2.33 \pm 0.14$
Crossing(%): $0.01 \pm 0.00$

(c) Full Fourier feature
$D_{W;1}$: $0.06 \pm 0.02$
Crossing(%): $0.01 \pm 0.00$

Figure 4: Results of quantile regression. $D_{W;1}$ is the Wasserstein 1-distance between ground truth quantile, and Crossing is the ratio of the numbers of crossing quantiles (%).

Figure 4 demonstrates the importance of the Fourier feature network in quantile regression. Here the networks are trained to estimate $F^{-1}_{\mathcal{N}(\mu,\sigma^2)}$ given $\mu \sim \mathcal{N}(0,1)$, $\log \sigma \sim \mathcal{N}(0,1)$. (a) is the result of a network trained by Vanilla IQN. (b) is the case when we handle the spectral bias in the level of quantile. (c) shows the case when we handle the spectral bias of all input domains. As shown, the spectral bias is the cause of inaccurate distribution estimation. Note that applying Fourier features in the

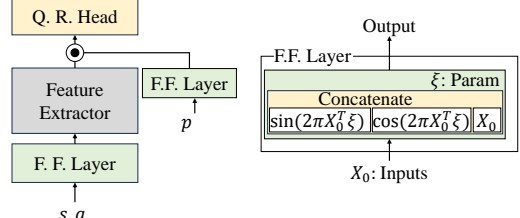

Figure 5: Fourier feature quantile regression network. $\odot$ is pairwise multiplication operator.

level domain[3] helps reduce the crossing quantile, but it does not improve overall accuracy. This is because the cosine embedding used in IQN already functions as a type of Fourier feature network. Figure 5 depicts the implementation of the Fourier feature network in (c). For parameters $\xi$ initialized from $\mathcal{N}(0, \sigma_{ff})$ and input $X_0$, the Fourier feature is

$$X_1 = \sin(2\pi X_0^T \xi) \# \cos(2\pi X_0^T \xi) \# X_0, \tag{21}$$

where $\#$ denotes the concatenation operator, and $X_1$ denotes the feature passed to the next layer. We also leverage using the Fourier feature IQN instead of cosine embedding IQN to reduce crossing quantile error. The remaining process is the same as conventional MLP.

## 5 EXPERIMENTS

### 5.1 BASELINES AND CRITERION

**Risk Measures.** For the experiments, we are interested in following risk measures.

**(i) CV@R.** considers average over below $\alpha$'s quantile. The risk measure's density $\phi^{\text{CV@R}}_\alpha$ parameterized by $\alpha \in [0,1]$ of is

$$\phi^{CV@R}_\alpha(p) = \frac{1}{\alpha} \mathbf{1}_{[0,\alpha]}(p). \tag{22}$$

**(ii) Wang.** is a risk measure motivated by areas where one's cost becomes another's benefit, like an option trading (Wang, 2002). The density of *Wang* risk measure, $\phi^{\text{wang}}_\alpha$, parameterized by $\alpha < 0$ is

$$\phi^{wang}_\alpha(p) = \frac{\varphi(F^{-1}_{\mathcal{N}(0,1)}(1-p) + \alpha)}{\varphi(F^{-1}_{\mathcal{N}(0,1)}(1-p))}, \tag{23}$$

where $F^{-1}_{\mathcal{N}(0,1)}$ and $\varphi$ are the quantile and density of standard normal distribution, respectively.

---

[3]The input domain of quantile function; i.e., the space of $p$ in $F^{-1}_Z(p)$.

**Baseline Algorithms.** ORAAC (Urpí et al., 2021) is a risk-sensitive variant of BCQ (Fujimoto et al., 2019). It achieves the risk-sensitive offline RL objective by only having slight modifications in the dataset's actions. **CODAC** (Ma et al., 2021) is a risk-sensitive variant of CQL (Kumar et al., 2020). It achieves a dataset constraint by minimizing KL-divergence between the RL agent's actions and the dataset's actions using the Donsker-Varadhan representation. **IQN-TD3+BC** is a risk-sensitive variant of the TD3+BC (Fujimoto & Gu, 2021) algorithm, where the IQN-Critic is used. The optimal actor $\pi_{\text{TD3+BC}}$ in IQN-TD3+BC is defined as

$$\pi_{\text{TD3+BC}} = \arg\min_{\pi} \lambda_{\text{q.learn}} H_{\phi}(Z^{\pi}(s,a)) + (a - a_{\mathcal{D}})^2. \tag{24}$$

Here $\lambda_{\text{q.learn}} > 0$ is a scale hyperparameter, $a_{\mathcal{D}}$ is a batch action, same as in Eq.(11).

## 5.2 MAIN RESULTS

We evaluate MR-IQN on finance and self-driving scenarios comparing with the baselines above. The label Mean is the average of the mean score over seeds and $-H_{\phi}(Z^{\pi})$ is the mean negative risk over seeds. All reported scores are averaged across 5 seeds. Numerical descriptions provided without explicit criteria are the results of comparing the negative risk. We also present the D4RL results, comparing the baselines and 1R2R, in Appendix A.1. The results and analysis about $CV@R(10\%)$ risks are in Appendix A.4, and the dataset details are provided in Appendix B.

Table 1: Performance comparison in finance scenarios.

| Target Risk Measure | | CV@R (50%) | | Wang (−0.5) | |
|---|---|---|---|---|---|
| Env. | Algorithm | Mean | $-H_{\phi}(Z^{\pi})$ | Mean | $-H_{\phi}(Z^{\pi})$ |
| Forex Env. (High) | CODAC | $1577.47 \pm 41.52$ | $249.53 \pm 28.33$ | $1591.76 \pm 8.12$ | $892.34 \pm 54.06$ |
| | ORAAC | $302.37 \pm 19.27$ | $-99.90 \pm 0.01$ | $304.89 \pm 38.05$ | $70.47 \pm 17.22$ |
| | IQN-TD3+BC | $2103.16 \pm 103.43$ | $454.16 \pm 11.76$ | $2181.16 \pm 123.41$ | $1307.52 \pm 90.05$ |
| | MR-IQN (ours) | $\mathbf{2537.52 \pm 289.57}$ | $\mathbf{556.94 \pm 172.92}$ | $\mathbf{2681.40 \pm 114.00}$ | $\mathbf{1639.79 \pm 57.88}$ |
| Stock Env. (Low) | CODAC | $37.67 \pm 7.54$ | $13.98 \pm 2.34$ | $33.43 \pm 7.01$ | $21.89 \pm 4.65$ |
| | ORAAC | $\mathbf{78.48 \pm 3.27}$ | $26.07 \pm 2.25$ | $78.57 \pm 6.09$ | $50.97 \pm 3.98$ |
| | IQN-TD3+BC | $\mathbf{80.49 \pm 1.87}$ | $\mathbf{30.92 \pm 1.15}$ | $83.10 \pm 3.05$ | $55.55 \pm 1.76$ |
| | MR-IQN (ours) | $74.78 \pm 2.10$ | $\mathbf{31.16 \pm 1.02}$ | $\mathbf{87.03 \pm 2.62}$ | $\mathbf{58.56 \pm 1.61}$ |

**Finance.** To investigate the real-world scenarios, we conduct experiments using the trading log data ranging from 2023-Feb-3rd to 2023-December-1st. The trading environment is based on the Meta-Trader Simulator (Amin, 2021). We evaluate 1000 episodes for each seed to calculate $CV@R$ and *Wang* negative risk. For the details, see Appendix. Table 1 depicts the results.

In the experiment, MR-IQN shows the highest performance in both Forex and Stock environments. In Forex environment, where uncertainty is high due to high leverage (High), the model risk approach effectively covers tail risk, maintaining high average returns and low risk, compared to the baseline approaches without model risk. IQN-TD3+BC, which is comparable to MR-IQN shows degraded performance (18.5%-11.2%) in Forex, indicating that MR-IQN is particularly useful in handling higher uncertainty and errors. Meanwhile, MR-IQN shows a slightly degraded average compared to ORAAC and IQN-TD3+BC in Stock environment. This indicates that there is a trade-off between conservatism and average performance.

**Self-Driving.** To simulate the self-driving scenario, we utilize the Airsim environment (Shah et al., 2017). In this scenario, RL agents are tasked with driving a quadcopter drone to a designated goal while avoiding obstacles, with wind disturbances affecting the drone's movement. The agents receive rewards based on the cosine of the angle between their movement direction and the goal. We evaluate performance in 100 episodes for each seed to measure negative risk. Table 2 depicts the results.

MR-IQN outperforms other baselines, showing the highest success rate (87.4%-88.8%) and the lowest collision rate (10.6%-11.6%). CODAC showed a relatively low average of rewards and a higher success rate compared to ORAAC. This implies that CODAC fails to generalize efficient actions and hovers over. The high collision rate of ORAAC implies that it fails to avoid obstacles because of the strict constraint to the batch action.

Table 2: Performance comparison in self-driving scenarios

| Risk Measure | Algorithm | Mean | $-H_\phi(Z^\pi)$ | Success(%) | Collision(%) |
|---|---|---|---|---|---|
| *CV@R* (50%) | CODAC | $43.56 \pm 89.45$ | $-425.72 \pm 97.73$ | $65.0 \pm 13.1$ | $21.4 \pm 5.5$ |
| | ORAAC | $92.37 \pm 40.51$ | $-88.37 \pm 4.20$ | $23.0 \pm 15.7$ | $68.0 \pm 10.2$ |
| | IQN-TD3+BC | $\mathbf{411.78 \pm 33.10}$ | $\mathbf{265.51 \pm 45.25}$ | $81.8 \pm 5.6$ | $\mathbf{14.8 \pm 5.6}$ |
| | MR-IQN(ours) | $\mathbf{431.44 \pm 30.41}$ | $\mathbf{290.96 \pm 46.95}$ | $\mathbf{88.0 \pm 4.6}$ | $\mathbf{11.6 \pm 4.9}$ |
| *Wang* (−0.5) | CODAC | $6.67 \pm 1.02$ | $-2.15 \pm 0.75$ | $52.6 \pm 19.3$ | $31.2 \pm 27.6$ |
| | ORAAC | $156.22 \pm 32.09$ | $51.73 \pm 27.53$ | $36.2 \pm 6.4$ | $58.4 \pm 8.1$ |
| | IQN-TD3+BC | $\mathbf{412.98 \pm 26.68}$ | $311.20 \pm 31.03$ | $\mathbf{83.8 \pm 5.1}$ | $14.4 \pm 3.9$ |
| | MR-IQN(ours) | $\mathbf{436.17 \pm 19.16}$ | $\mathbf{341.26 \pm 18.36}$ | $\mathbf{87.4 \pm 1.8}$ | $\mathbf{10.6 \pm 2.6}$ |

## 5.3 ABLATION STUDY

Table 3: Ablation results. ✓ represents presence and ✗ represents absence. M.R. and F.F. are the abbreviations for model risk and Fourier feature, respectively.

| Components | | | *CV@R* (50%) | | *Wang* (−0.5) | |
|---|---|---|---|---|---|---|
| M.R. | F.F. | TQC | Mean | $-H_\phi(Z^\pi)$ | Mean | $-H_\phi(Z^\pi)$ |
| ✓ | ✓ | ✓ | $\mathbf{2537.52 \pm 289.57}$ | $\mathbf{556.94 \pm 172.92}$ | $\mathbf{2681.40 \pm 114.00}$ | $\mathbf{1639.79 \pm 57.88}$ |
| ✓ | ✓ | ✗ | $2003.37 \pm 214.76$ | $464.84 \pm 57.01$ | $1975.85 \pm 104.49$ | $1160.65 \pm 87.82$ |
| ✓ | ✗ | ✓ | $\mathbf{2410.83 \pm 86.94}$ | $\mathbf{513.02 \pm 77.54}$ | $2324.31 \pm 146.95$ | $1376.27 \pm 122.96$ |
| ✓ | ✗ | ✗ | $1763.21 \pm 278.16$ | $278.16 \pm 94.26$ | $1607.13 \pm 76.54$ | $872.54 \pm 56.23$ |
| ✗ | ✓ | ✓ | $2103.16 \pm 103.43$ | $454.16 \pm 11.76$ | $2181.16 \pm 123.41$ | $1307.52 \pm 90.05$ |
| ✗ | ✓ | ✗ | $1712.79 \pm 75.40$ | $295.10 \pm 49.83$ | $1650.28 \pm 51.48$ | $934.79 \pm 30.00$ |
| ✗ | ✗ | ✓ | $2170.68 \pm 186.55$ | $476.23 \pm 75.94$ | $2188.57 \pm 134.40$ | $1297.15 \pm 88.15$ |
| ✗ | ✗ | ✗ | $1454.30 \pm 76.87$ | $153.84 \pm 38.57$ | $1414.44 \pm 40.00$ | $750.22 \pm 37.45$ |

We also conduct an ablation study in the Forex environment, identifying model risk, Fourier features, and TQC as key contributors to the model's improvement. Table 3 presents the results. MR-IQN achieves an 8.6%-19.8% performance gain from the Fourier feature and a 19.8%-41.3% performance gain from TQC. The performance gain from model risk is reported in Section 5.2. Notably, model risk enhances performance when combined with other factors, due to the design of the critic ensemble criterion and the requirement for accurate estimation of $\mu$ and $\sigma$, as discussed in Section 4.2.

Our framework achieves approximately 118.6%-262.0% performance gains compared to the naive implementation, which omits the key components. The reason why TQC is introduced to our framework is to construct a target quantile which is similar to $F_{\text{ens.}}^{-1}$. As a result, the calculated model risk tolerance, $\varepsilon$, becomes more aligned with our intention, i.e., large $\varepsilon$ for high inconsistency and vice versa. However, the performance improvement of TQC without other factors (72.9%-209.6%) implies that TQC itself is not a negligible factor.

## 6 CONCLUSION

We proposed a model risk-sensitive offline RL framework, devising the critic-ensemble criterion to capture model risk effectively. To ensure the precision of model risk calculation, we employed Fourier feature networks, which accurately estimate both the mean and standard deviation—critical components for calculating the model risk. The framework ensures more reliable decision-making in risk-sensitive applications by accounting for potential model errors and striving to make the best decisions despite them. While our framework is limited to spectral risk measures and cannot accommodate those outside this class, such as *CMV* (Vadori et al., 2020) or *EV@R* (Ni & Lai, 2022), the broad applicability of spectral risk measures covers many practical real-world problems. Our future work involves extending the framework to partially observable Markov decision processes, including embodied control and decision-making in mission-critical business applications.

## ACKNOWLEDGEMENT

This work was supported by Institute of Information & communications Technology Planning & Evaluation (IITP) grant funded by the Korea government (MSIT), ( RS-2022-II220043 (2022-0-00043), Adaptive Personality for Intelligent Agents, RS-2022-II221045 (2022-0-01045), Self-directed multi-modal Intelligence for solving unknown, open domain problems, RS-2025-02218768, Accelerated Insight Reasoning via Continual Learning, and RS-2019-II190421, Artificial Intelligence Graduate School Program (Sungkyunkwan University)), the National Research Foundation of Korea (NRF) grant funded by the Korea government (MSIT) (No. RS-2023-00213118), IITP-ITRC (Information Technology Research Center) grant funded by the Korea government (MIST) (IITP-2025-RS-2024-00437633, 10%), IITP-ICT Creative Consilience Program grant funded by the Korea government (MSIT) (IITP-2025-RS-2020-II201821, 10%), and by Samsung Electronics.

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

# APPENDIX

## A ADDITIONAL EXPERIMENTS

### A.1 D4RL

Since D4RL is a standard benchmark for offline RL training, we report the D4RL scores here. We trained MR-IQN using the *Wang* risk measure with a parameter of $-0.5$. MR-IQN shows comparable performance (68.59) to the state of the art risk-sensitive offline RL algorithm 1R2R (67.35-67.78) in overall scores and outperforms it when evaluating the average without random dataset (85.44), which is a metric often used in behavior clone based offline RL algorithms (Wang et al., 2023; Lu et al., 2023).

Table A: D4RL Mujoco-v2 results (normalized score).

| | Environment | MR-IQN$_{Wang-0.5}$ | 1R2R$_{Wang}$ | 1R2R$_{CV@R}$ | ORAAC | CODAC | TD3+BC |
|---|---|---|---|---|---|---|---|
| Random | HalfCheetah | $20.48 \pm 0.91$ | 35.5 | **36.9** | 22.4 | 31.0 | 11.0 |
| | Hopper | $26.47 \pm 10.6$ | **31.8** | 30.9 | 10.9 | 9.2 | 8.5 |
| | Walker2d | $7.15 \pm 1.43$ | 8.1 | 7.6 | 2.4 | **12.8** | 1.6 |
| Medium | HalfCheetah | $61.14 \pm 0.75$ | **75.5** | **74.5** | 38.5 | 62.8 | 48.3 |
| | Hopper | $\mathbf{98.84 \pm 5.87}$ | 64.6 | 80.2 | 29.6 | 63.9 | 59.3 |
| | Walker2d | $\mathbf{89.05 \pm 1.24}$ | 83.4 | 63.9 | 45.5 | 84.0 | 83.7 |
| Medium Replay | HalfCheetah | $46.36 \pm 2.19$ | **65.6** | **65.7** | 39.3 | 53.4 | 44.6 |
| | Hopper | $\mathbf{95.08 \pm 7.14}$ | **93.2** | 92.9 | 22.6 | 68.8 | 60.94 |
| | Walker2d | $81.07 \pm 13.93$ | 88.4 | **92.2** | 11.1 | 73.9 | 81.8 |
| Medium Expert | HalfCheetah | $78.54 \pm 12.00$ | **94.5** | **96.0** | 24.4 | 76.7 | 90.7 |
| | Hopper | $\mathbf{105.62 \pm 3.60}$ | 89.6 | 81.6 | 4.1 | 87.6 | 98.0 |
| | Walker2d | $\mathbf{113.25 \pm 0.36}$ | 78.1 | 90.9 | 42.8 | **112.0** | **110.1** |
| | Average | **68.59** | **67.35** | **67.78** | 31.83 | 61.34 | 58.86 |
| | Average without Random | **85.44** | 81.43 | 81.99 | 36.08 | 75.90 | 76.13 |

### A.2 BEHAVIOR ANALYSIS

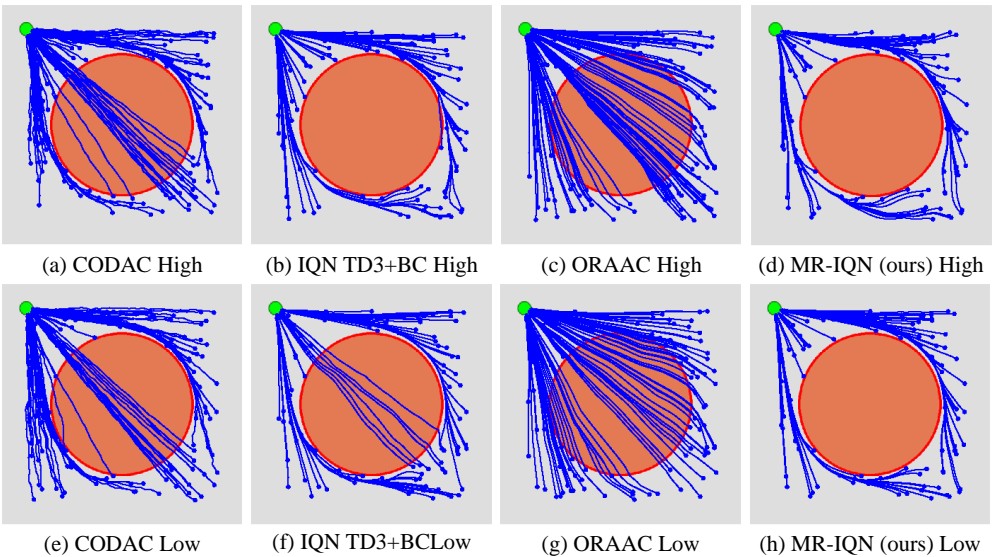

(a) CODAC High    (b) IQN TD3+BC High    (c) ORAAC High    (d) MR-IQN (ours) High

(e) CODAC Low    (f) IQN TD3+BCLow    (g) ORAAC Low    (h) MR-IQN (ours) Low

Figure A: Risky point mass environment. In the red zone, the agent gets a random reward. Top (a-d) figures depict the behavior of the policies when the variance of reward is high. Bottom (e-h) figures depict the behavior of the policies when the variance of reward is low. The blue lines are the trajectories of policies.

To analyze the behavior of risk-sensitive offline RL algorithms, we compare the algorithms in a modified risky point mass environment (Ma et al., 2021). The agent is trained to reach the goal (green points in Figure A). In this environment, there is a red zone, where the agent gets normal white noise in the reward signal in $0.1$ probability. We compare two situations: when the variance of the noise signal is high ($100$) and low ($5$). Figure A depicts the results.

In this environment, MR-IQN (ours) shows the most risk-averse behavior, avoiding the red zone as much as possible, as desired. IQN TD3+BC sometimes fails to avoid the red zone, especially when the variance of the reward signal is low. This difference highlights the conservatism of model risk-sensitive offline RL compared to conventional risk-sensitive offline RL. In contrast, both CODAC and ORAAC consistently fail to avoid the red zone. CODAC's failure is due to the key assumption of strict monotonicity of quantile function (Ma et al., 2021), which does not hold in this environment, coupled with hyperparameter sensitivity inherited from CQL (Tarasov et al., 2024). ORAAC struggles because the strong constraints imposed by the behavior policy make it difficult for the policy to stitch together the desired trajectory.

## A.3  NUMBER OF CRITICS

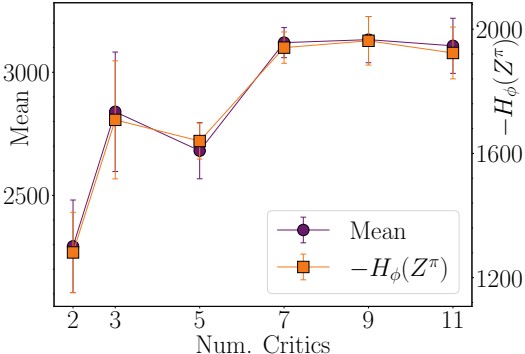

Figure B: Effect of the number of critics (i.e., ensemble size). The environment is Forex and the risk measure is *Wang*(-0.5).

We also investigate how the number of critics affects performance. Figure B shows the results. As demonstrated in TQC Kuznetsov et al. (2020), the performance improvement begins to saturate between 5 and 7 critics for both the mean and risk performance. The observed performance degradation between 3 and 5 critics appears to be an illusion caused by the presence of outliers, as suggested by the large gaps in the error bars.

## A.4  TAIL SCORE ANALYSIS

Table B: *CV@R*(10%) Performance in finance scenarios

|  | Target Risk Measure | CODAC | ORAAC | IQN-TD3+BC | MR-IQN (ours) |
|---|---|---|---|---|---|
| Stock Env. | *Wang*(-0.5) | $3.00 \pm 0.91$ | $7.92 \pm 0.86$ | $8.32 \pm 0.71$ | $\mathbf{10.14 \pm 0.55}$ |
|  | *CV@R*(50%) | $2.71 \pm 1.51$ | $7.85 \pm 0.82$ | $8.33 \pm 0.67$ | $\mathbf{10.06 \pm 0.53}$ |
|  | *CV@R*(10%) | $3.98 \pm 0.86$ | $7.91 \pm 0.74$ | $8.68 \pm 0.43$ | $\mathbf{10.35 \pm 0.52}$ |
| Forex Env. | *Wang*(-0.5) | $-51.27 \pm 22.79$ | $-96.59 \pm 1.04$ | $-27.91 \pm 48.02$ | $9.44 \pm 59.11$ |
|  | *CV@R*(50%) | $-48.72 \pm 25.25$ | $-100 \pm 0.00$ | $-11.47 \pm 28.97$ | $25.71 \pm 32.09$ |
|  | *CV@R*(10%) | $-4.35 \pm 2.06$ | $-47.54 \pm 20.94$ | $0.25 \pm 34.37$ | $\mathbf{31.40 \pm 36.02}$ |

We further investigated the *CV@R*(10%) (negative) risk in financial scenarios. The results are presented in Table B. In Stock environment, only CODAC demonstrates a significant improvement in *CV@R*(10%) risk, whereas the performance of other algorithms appears to have reached saturation.

In Forex environment, all algorithms exhibit consistent behavior, with $CV@R(10\%)$ optimization yielding the highest $CV@R(10\%)$ performance, followed by $CV@R(50\%)$ and $Wang(-0.5)$ optimization criteria, respectively, except for ORRAC, which does not show meaningful $CV@R(10\%)$ performance in either $CV@R(50\%)$ or $Wang(-0.5)$ risks.

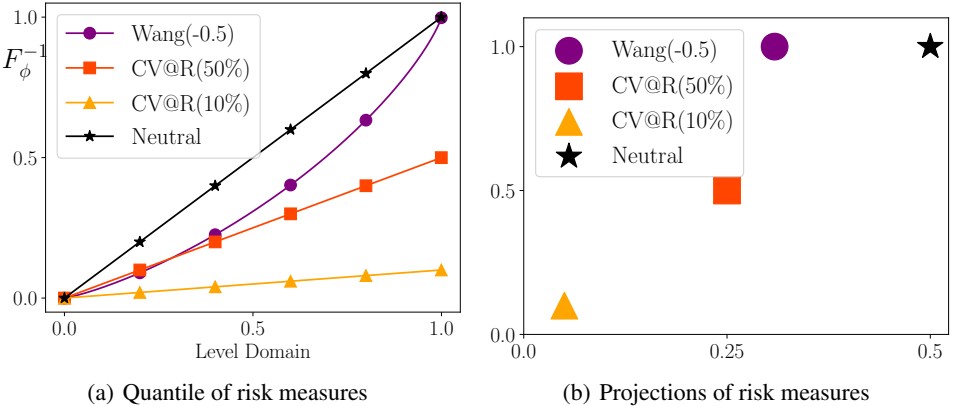

(a) Quantile of risk measures    (b) Projections of risk measures

Figure C: Visualizing risk measures

To examine this phenomenon, we investigate how risk measures are distinguishable. We leverage the Wasserstein-2 distance ($D_{W;2}$) to measure distinguishability. The $D_{W;2}$ between $CV@R(50\%)$ and $CV@R(10\%)$ is approximately 0.23, and $D_{W;2}$ between $Wang(-0.5)$ and $CV@R(10\%)$ is approximately 0.40. Therefore, $CV@R(50\%)$ is a more *similar* risk measure with $CV@R(10\%)$ compared to $Wang(-0.5)$. This similarity explains the consistent behavior of algorithms. Figure C visualizes the risk measures. Figure C (a) describes the quantile function of risk measures, and Figure C (b) describes the vector projection of risk measures via hat function coefficient by finite element method approximation. We can verify that the previous argument.

Note that spectral risk measure is defined as (Eq.(3)),

$$H_\phi(Z) = -\int_0^1 F_Z^{-1}(p)\phi(p)dp.$$

The equation is governed by $\phi$, and $\phi$ is a probability density function defined on $[0, 1]$. Therefore, it is natural to use Wasserstein distance to measure distinguishability. Meanwhile, the model risk calculates the risk of some function whose $D_{W;2}$ distance between reference distribution is constrained by $\sqrt{\varepsilon}$ as defined in Eq.(9). Therefore, it shows similar behavior to other algorithms.

## B  EXPERIMENT DETAILS

### B.1  DETAILED EXPLANATION OF BASELINES

**ORAAC** policy $\pi_{\text{oraac}}$ is as follows:

$$\pi_{\text{oraac}} = \pi_{\text{bc}}(s) + \kappa\pi_{\text{risk}}(s, \pi_{\text{bc}}(s)) \tag{A}$$

where $\pi_{\text{bc}}$ is behavior cloning policy implemented using VAE, $\pi_{\text{risk}}$ is the risk-sensitive perturbation policy, and $\kappa \geq 0$ is the modification parameter. We set $\kappa = 0.25$ following Urpí's base implementation.

**CODAC**'s critic objective is as follows

$$Z_{\text{CODAC}}^* = \arg\min_Z \max_\pi \mathcal{L}_{\text{crit.}}(\theta)+$$

$$\lambda \underset{p\sim U[0,1]}{\mathbb{E}} \underbrace{\left( \underset{a\sim\mathcal{D}}{\mathbb{E}}[F_Z^{-1}(p;s,a,\theta)] - \log\underset{a\sim\pi}{\mathbb{E}}[\exp(F_Z^{-1}(p;s,a,\theta))] \right)}_{\text{KL-potential by Donsker-Varadhan Representation}}. \tag{B}$$

Table C: Performance of ORAAC and CODAC with/without observation normalization.

| Target Risk Measure | | *CV@R* (50%) | | *Wang* (-0.5) | |
|---|---|---|---|---|---|
| Without Observation Normalization | | | | | |
| Env. | Algorithm | Mean | $-H_\phi(Z^\pi)$ | Mean | $-H_\phi(Z^\pi)$ |
| Forex | CODAC | $-0.01 \pm 0.04$ | $-0.03 \pm 0.09$ | $3.63 \pm 8.12$ | $-1.74 \pm 3.52$ |
| | ORAAC | $302.37 \pm 19.27$ | $-99.90 \pm 0.01$ | $304.89 \pm 38.05$ | $70.47 \pm 17.22$ |
| Stock | CODAC | $6.67 \pm 1.02$ | $-2.15 \pm 0.75$ | $5.89 \pm 1.64$ | $0.48 \pm 1.40$ |
| | ORAAC | $68.6 \pm 13.0$ | $24.0 \pm 3.59$ | $73.43 \pm 3.43$ | $47.68 \pm 2.21$ |
| Airsim | CODAC | $-4190.24 \pm 3073.29$ | $-7460.17 \pm 5227.62$ | $6.67 \pm 1.02$ | $-2.15 \pm 0.75$ |
| | ORAAC | $-149.58 \pm 64.53$ | $-431.80 \pm 217.92$ | $68.6 \pm 13.0$ | $24.0 \pm 3.59$ |
| With Observation Normalization | | | | | |
| Forex | CODAC | $1577.47 \pm 41.52$ | $249.53 \pm 28.33$ | $1591.76 \pm 8.12$ | $892.34 \pm 54.06$ |
| | ORAAC | $0.00 \pm 0.00$ | $0.00 \pm 0.00$ | $0.00 \pm 0.00$ | $0.00 \pm 0.00$ |
| Stock | CODAC | $37.67 \pm 7.54$ | $13.98 \pm 2.34$ | $33.43 \pm 7.01$ | $21.89 \pm 4.65$ |
| | ORAAC | $78.48 \pm 3.27$ | $26.07 \pm 2.25$ | $78.57 \pm 6.09$ | $50.97 \pm 3.98$ |
| Airsim | CODAC | $43.56 \pm 89.45$ | $-425.72 \pm 97.73$ | $-2.40 \pm 17.59$ | $-301.23 \pm 109.30$ |
| | ORAAC | $92.37 \pm 40.51$ | $-88.37 \pm 4.20$ | $156.22 \pm 32.09$ | $51.73 \pm 27.53$ |
| Behavior Policy | | | | | |
| Forex | | 191.42 | $-99.90$ | - | 26.91 |
| Stock | | 1.072 | $-11.67$ | - | $-6.85$ |
| Airsim | | 378.93 | 78.57 | - | 188.29 |

Here $\mathcal{L}_{\text{crit.}}$ is the quantile regression loss with respect to distributional Bellman residuals defined in Eq.(4), and $\lambda > 0$ is constant or lagrangian coefficient. Note that the right most part is equal to learning f-GAN discriminator (Nowozin et al., 2016) with KL-divergence (Nguyen et al., 2010). Therefore, CODAC (and CQL) can be understood as GAN + RL.

This leads the optimal actor $\pi_{\text{codac}}$ is defined as

$$\pi_{\text{codac}} = \arg\min_\pi H_\phi(Z^*_{\text{CODAC}}) = \arg\min H_\phi(Z^\pi(s,a)), \text{ constrained } D_{KL}(\pi_{\text{codac}} \| \pi_D) < \epsilon. \quad \text{(C)}$$

for some $\epsilon > 0$ and behavior policy $\pi_\mathcal{D}$ in Eq.(7). See Eq.(8) as well.

**1R2R** (Rigter et al., 2024) is a model-based risk-sensitive offline RL algorithm based on the dual formula for coherent[1] risk measure as Eq.(D).

$$H_\phi(Z^\pi(s,a)) = -\inf \mathbb{E}_{\mathbb{Q} \in \mathcal{T}(\phi}[Z^\pi(s,a)], \quad \text{(D)}$$

where $\mathcal{T}$ is a subset of probability measure which is absolutely continuous with respect to occupation measure of policy, $\rho^\pi$. The constrains of $\mathcal{T}$ is $\phi$ dependent. For example, for $CV@R_\alpha$ risk measure, $\mathcal{T} = \{\mathbb{P} \ll \rho^\pi | \frac{d\mathbb{P}}{d\rho^\pi} < \frac{1}{\alpha}\}$ where $\ll$ denotes absolute continuity and $\frac{d\mathbb{P}}{d\rho^\pi}$ denotes the Radon-Nikodym derivative between $\mathbb{P}$ and $\rho^\pi$.

To calculate risk based on Eq.(D), 1R2R samples next states, $\{s_{t+1}^{(i)}\}_{i=0}^N$ given $s_t, a_t$, with Gaussian ensemble model. Then, construct empirical quantile function based on

$$F_{Z^\pi}^{-1}(p_i; s, a) \approx R(s_t, a_t) + \gamma V^\pi(s_{t+1}^{(i)}) \quad \text{(E)}$$

where $V^\pi(s_{t+1}^{(1)}) \leq V^\pi(s_{t+1}^{(2)}) \leq \cdots \leq V^\pi(s_{t+1}^{(N)})$ and $p_1 = 0, p_2 = \frac{1}{N-1}, \ldots, p_N = 1$.

According to the risk measured by approximated quantile function in Eq.(E), the synthetic data $(s_t, a_t, R(s,a), s_{t+1})$ is rejected or accepted; i.e., $s_{t+1}$ is resampled according to $\phi$. Therefore, 1R2R minimizes dynamic risk (time-dependent risk) instead of static risk (time-independent risk). However, 1R2R requires terminal function given observation a priori (Rigter et al., 2023). Therefore, we could not conduct the main experiments with 1R2R.

---

[1]The set of all coherent risk measures includes the set of spectral risk measures; i.e., every spectral risk measure is a coherent risk measure.

**ORAAC and CODAC Tuning:** To improve the performance of the baselines (ORAAC and CO-DAC), we also apply observation normalization. The following tables present baseline performance both with and without observation normalization, and in the main text, we report the better score between them.

### B.2 ENVIRONMENT DETAILS

**Finance.** The trading environment is implemented using MT-sim (Amin, 2021). The data ranges from February 3rd to December 1st, 2023. In the Forex environment, the agent trades 'EURUSD' and 'USDJPY' with a leverage of 100, while in the Stock environment, it trades 'QQQ', 'SPY', 'IWM', and 'IYY' with a leverage of 10. The forex data is collected from MetaTrader 4 (Sajedi, 2024), and the stock data from Yahoo Finance (Perlin, 2023).

The environment terminates when the agent's assets drop to $0. To increase randomness, trades are initiated at a random interval, and the agent aims to maximize return within this interval. Observations include balance, equity, margin, orders, and features. The features consist of 15-day accumulated price and volume data, including open, close, low, and high prices. For numerical stability, the data is preprocessed using the arcsinh function, achieving logarithmic scaling. Action represents logits for buying, selling, or holding each symbol. The reward is the benefit ratio relative to the initial balance, fixed at $50,000.

**Self-driving.** The self-driving environment is implemented using Airsim (Shah et al., 2017). The agent's goal is to pilot a quadcopter drone to the target while avoiding static obstacles with height constraints. Random walk noise distorts the agent's actions to simulate wind conditions, introducing randomness. The environment ends when the drone collides with an object or reaches the goal.

The observations provided to the agent include lidar data, previous wind conditions, kinematic information (e.g., orientation, linear velocity), and the goal position. The lidar data has a fixed array shape as it is projected onto a grid in the spherical coordinate system. For each grid cell, the minimal value of radii is taken, capturing the closest detected object in that region. The agent's actions represent the pulse width modification (PWM) values of the motors which corresponds to the RPM of motors. The reward is determined by the cosine similarity between the drone's linear velocity and the goal position. Further, the agent gets $100$ reward if it reaches the goal and $-100$ if it collides. Further, the agent receives alive reward of $0.01$ when the agent does not collide with anything. To avoid trivial solution, the agent receives height constrain reward when if flies to high; $-\max(\text{current height} - \text{height constrain}, 0)$.

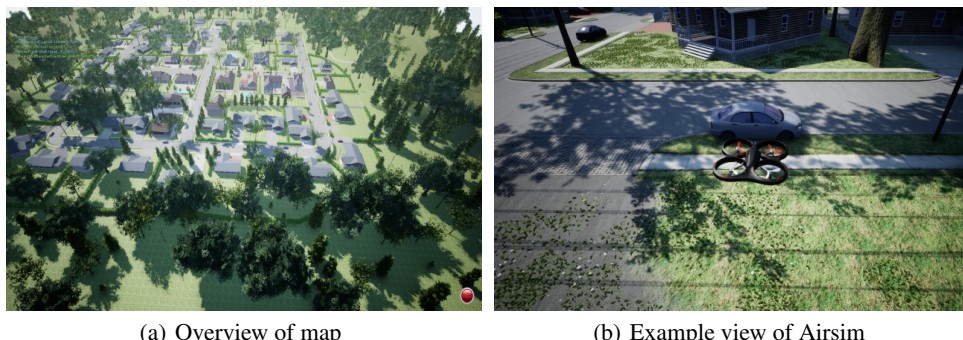

(a) Overview of map          (b) Example view of Airsim

Figure D: Airsim environment

### B.3 HYPERPARAMETER DETAILS

Table D lists up the detailed hyperparameters of the experiments.

Table D: Hyperparameter settings

| Hyperparameter (Common) | Finance | Self-Driving | D4RL |
|---|---|---|---|
| learning rate | | $3 \times 10^{-4}$ | |
| optimizer | | Adam ($\beta_1 = 0.9, \beta_2 = 0.999$) | |
| discount factor ($\gamma$) | 0.99 | 0.96 | 0.99 |
| batch size | | 256 | |
| layer normalization | | True | |
| num. critics | 5 | 5 | 3 |
| num. quantiles for critic training | | 16 | |
| soft update ratio ($\tau$) | | $5 \times 10^{-3}$ | |
| IQN num. cosine (Fourier feature dim for $p$) | | 64 | |
| IQN embedding dim. | | 64 | |
| **Model** | | MR(ours) & IQN-TD3+BC | |
| q-learning scale ($\lambda_{q.\ learn}$) | 2.5 | 1.0 | 5.0 |
| Fourier feature init. ($\sigma_{ff}$) | | $1 \times 10^{-3}$ | |
| Fourier feature dimensions for $s, a$ | | 250 | |
| TQC drop out per critic | | 3 | |
| num. quantiles for actor train ($N$) | | 100 | |
| policy delay | | 2 | |
| **Model** | | ORAAC | |
| num. actions for inference | | 100 | |
| action distortion scale ($\kappa$) | | 0.25 | |
| VAE KL-divergence coef. ($\beta$) | | 0.5 | |
| **Model** | | CODAC | |
| KL-Lagrangian | | True | |
| KL-Lagrangian target | | 10 | |
| entropy coef | | 0.2 | |

## C  MINOR COMMENTS ABOUT TERMINOLOGY

In the actuarial science literature, conditional value at risk (*CV@R*), tail value at risk (*TV@R*), expected shortfall (*ES*), and conditional tail expectation (*CTE*) are more strictly distinguished. However, in the main text, we use *CV@R* to mean *TV@R*, following the ML literature. These terms must be distinguished when the quantile function of a random variable is discontinuous at the confidence level. Because the worst-case quantile function for calculating model risk might also be discontinuous at that level, we acknowledge that our usage actually refers to *TV@R* rather than *CV@R*, though we keep using *CV@R* for consistency with previous ML literature. For details on their differences, see Zhou et al. (2015).

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
