# OpenReview forum: "Model Risk-sensitive Offline Reinforcement Learning"
_ICLR.cc/2025/Conference — ICLR 2025 Poster_

### Official Review · Reviewer_3isb · 2024-10-25

**Soundness:** 2
**Presentation:** 2
**Contribution:** 2
**Rating:** 6
**Confidence:** 3

**Summary:**

This work considers the risk-sensitive setting in offline RL, which is important for real-world applications. This work proposes to minimize the worst-case risks across a set of plausible alternative scenarios rather than solely focusing on minimizing estimated risk. The proposed method, MR-IQN, introduces a critic-ensemble criterion method, the learned Fourier feature framework, and the IQN framework to address spectral bias in neural networks. Extensive experiments show that MR-IQN can significantly reduce the different kinds of risk, like CVaR and Wang risk.

**Strengths:**

- Considering spectral risk measure is interesting in risk-sensitive RL as it seems a general form of several risk metrics like CVaR, worst case, and Wang risk.

- Extensive ablation studies show the importance of each component in MR-IQN.

- Fig. 4 explains the advantage of Fourier feature clearly.

**Weaknesses:**

- It seems that the algorithm is a direct application of previous theoretical results (Bernard et al., 2023) as the major theoretical results (Thm 1, Cor 1) are from (Bernard et al., 2023). What is the major challenge in applying these theoretical analyses to your methods?

- There seems to be a gap between the objective 10 (can be simplified by Cor. 1) and the proposed algorithm. Why can MR-IQN handle objective 10? As objective 10 is a constrained optimization problem and hard to solve directly, do you use bi-level optimization, or use the Lagrange multiplier method, or some other methods? It will make the work more solid by providing a more detailed explanation of how MR-IQN addresses the constrained optimization problem in objective 10, including any specific optimization techniques they use.

- Lack of some related work on several risk-sensitive RL methods including different risk metrics which are closely related to this work, like CVaR [1-4], EaR [5], worst case [6], and so on.

Reference:

[1] TRC: Trust region conditional value at risk for safe reinforcement learning

[2] Towards safe reinforcement learning via constraining conditional value-at-risk

[3] Distributional reinforcement learning for risk-sensitive policies

[4] Efficient risk-averse reinforcement learning

[5] Risk-sensitive reinforcement learning via Entropic-VaR optimization

[6] Provably efficient risk-sensitive reinforcement learning: Iterated cvar and worst path

**Questions:**

- Why can we translate the objective (8) to the objective (10)?

- In step 1, why does MR-IQN choose the **smallest** expectation and the **largest** deviation?

- Why do Fourier features benefit a lot, are there any explanations?

- Are there ablation studies about the number of ensembled critics?

---

> ### Author Response · Authors · 2024-11-17
> **Weakness**
>
> We appreciate your meaningful feedback.
>
> ### **Weakness 1. Major challenge to apply the Theoretical Results of Bernard et al.'s work**
> The main challenge lies in the assumption of Bernard et al.'s work, which is somewhat unrealistic in an RL setting and not well-suited as a formula for machine learning, given its origins in financial mathematics.
>
> * Bernard et al.'s work requires exact ground truth values for the mean ($\mu$), deviation ($\sigma$), and a finely tuned tolerance ($\varepsilon$), which is impossible in offline RL because of "deadly triad" [1], "distribution shift" [2], and error correlation [3], and they prevent to learn the ground truth return distributions. We propose the method to provide a practical solution to deal with this problem (i.e., Fourier feature network) and propose a proper formula to calculate model risk so that its policy gradient is well aligned with the RL objective.
>
> * Since model risk is not designed for machine learning, it might fail in edge cases during training. Our framework minimizes these issues by carefully selecting $\mu$, $\sigma$, and $\varepsilon$ and by incorporating stop-gradient techniques and edge-case handling methods to ensure proper training signals, as demonstrated by our results.
>
> Reviewer DSTd Q.1 had a similar concern in Weakness 1. Please also refer to the response to the reviewer DSTd for details.
>
> [1] Van Hasselt, Hado, et al. "Deep reinforcement learning and the deadly triad." arXiv preprint arXiv:1812.02648 (2018).
>
>  [2] Fu, Justin, et al. "Diagnosing bottlenecks in deep q-learning algorithms." International Conference on Machine Learning. PMLR, 2019.
>
> [3] Fujimoto, Scott, et al. "Why Should I Trust You, Bellman? The Bellman Error is a Poor Replacement for Value Error." International Conference on Machine Learning. PMLR, 2022.
>
>
>
> ### **Weakness 2. Equation (10) requires constrained optimization**
>
>  Yes, the direct solution requires a Lagrangian multiplier, and some offline RL directly solve the constrained optimization by introducing a Lagrangian multiplier (e.g., CQL[4], DICE framework [5-6]). But many other offline RL approaches like TD3+BC [7], diffusion QL [8], BCQ [9], and decision transformer [10] solve the problem in an auxiliary way; i.e., *reverse engineering* Lagrangian multiplier.
>
> $$
> \pi^{\*}= \text{arg} \min \lambda\_{\text{q.learn}} \text{MR}\_{\phi}(Z^{\pi}(s,a); \mu, \sigma, \varepsilon) + (a - a\_{\mathcal{D}})^{2} =  \text{arg} \min \lambda\_{\text{q.learn}} \text{MR}\_{\phi}(Z^{\pi}(s,a); \mu, \sigma, \varepsilon) + D_{KL}(a \lVert  a\_{\mathcal{D}})
> $$
> $$
> = \text{arg} \min  \text{MR}\_{\phi}(Z^{\pi}(s,a); \mu, \sigma, \varepsilon) \text{ such that } D\_{KL}(a \lVert  a\_{\mathcal{D}}) < \epsilon({\lambda\_{\text{q.learn}}})
> $$
> The equation holds provided when the dataset is unimodal. Please note that all data don't have to be unimodal, but some of them are allowed to be multi-modally distributed. Please also note that the conventional Lagrangian multiplier method finds $\lambda$ for given $\epsilon$ by gradient descent.
>
> In this auxiliary method, we assume that $\epsilon$ is not explicitly given and consider $\epsilon$, which is determined by $\lambda_{\text{q.learn}}$ as some hyperparameter.
>
>
> Wang et al. [8] provided a well-organized overview of this taxonomy for dataset constraints in offline RL in their work.
>
> Reviewer AhNq had a similar concern. Please see the response to AhNq Weakness 1-2 for the details as well.
>
>
> [4] Kumar, Aviral, et al. "Conservative q-learning for offline reinforcement learning." Advances in Neural Information Processing Systems 33 (2020): 1179-1191.
>
> [5] Lee, Jongmin, et al. "COptiDICE: Offline Constrained Reinforcement Learning via Stationary Distribution Correction Estimation." International Conference on Learning Representations.
>
>
> [6] Nachum, Ofir, and Bo Dai. "Reinforcement learning via fenchel-rockafellar duality." arXiv preprint arXiv:2001.01866 (2020).
>
> [7]  Fujimoto, Scott, and Shixiang Shane Gu. "A minimalist approach to offline reinforcement learning." Advances in neural information processing systems 34 (2021): 20132-20145.
>
> [8] Wang, Zhendong, Jonathan J. Hunt, and Mingyuan Zhou. "Diffusion Policies as an Expressive Policy Class for Offline Reinforcement Learning." The Eleventh International Conference on Learning Representations.
>
> [9] Fujimoto, Scott, David Meger, and Doina Precup. "Off-policy deep reinforcement learning without exploration." International conference on machine learning. PMLR, 2019.
>
> [10] Chen, Lili, et al. "Decision transformer: Reinforcement learning via sequence modeling." Advances in neural information processing systems 34 (2021): 15084-15097.
>
> ### **Missing related works**
> We appreciate your recommendations and added the works to the related work section (revised version lines 109-116).

---

> ### Author Response · Authors · 2024-11-18
> **Questions**
>
> ### **Question 1. From objective (8) to objective (10)**
> The objective (10) is directly analogy to the objective (8). We want to minimize model risk instead of risk measured by critics. Therefore, we replaced the objective $H_{\phi}(Z^{\pi}(s, a))$ to the $\text{MR}(Z^{\pi}(s, a); \mu, \sigma, \varepsilon)$.
>
> Please recall that $\text{MR}(Z^{\pi}(s, a); \mu, \sigma, \varepsilon)$ accommodates cases where distribution shifts result in non-overlapping supports, as discussed in the related work section on model risk (and in response to Reviewer AhNq, Weakness 1: Theoretical Analysis). The cases often happen in offline RL domains.
>
> ### **Question 2. Choice of $\mu$ and $\sigma$**
>
> The main reasons are
> * Reducing overestimation bias in actor-critic methods (Fujimoto et al. [11])
> * Reducing the edge case, when $\varepsilon < (\mu_{Z} - \mu)^{2} + (\sigma_{Z} - \sigma)^{2}$ (equation in Theorem 1.) (i.e., when the model risk does not have a solution).
>
> **Reducing Overestimation Bias**: Fujimoto et al. [11] have shown that using minimal expected value ($\mu$) reduces the overestimation bias in online RL. The maximal standard deviation ($\sigma$) is an analogy of choosing minimal $\mu$, as large standard deviation $\sigma$ is something cost (i.e., similar to negative reward).
>
> **Reducing Edge Case**: As we choose the minimal $\mu$ and maximal $\sigma$ for calculating model risk, the edge case occurs when one of the $\mu$ or $\sigma$  is not chosen from the reference distribution, which is the most conservative critic as
>
> $$F^{-1}\_{\text{ref}}(\cdot) = F^{-1}\_{Z^{\pi}}(\cdot; s, a, \theta^{(\*)}), \text{ where } \theta^{(\*)} = \text{argmax}\_{\theta^{(i)}} H\_{\phi}(Z^{\pi}(s, a, \theta^{(i)})).$$
>
> However, since the reference distribution is the most conservative one, it implies that the reference distribution has the smallest $\mu$ and largest $\sigma$ in general. Except in the very early stage of the training, the probability both $\mu$ and $\sigma$ are chosen from the reference distribution is high, making $(\mu_{Z} - \mu)^{2} + (\sigma_{Z} - \sigma)^{2} = 0$, as all critics are trained with same objective function and TD-target. This prevents the edge case.
>
> Reviewer AhNq and DSTd asked a similar question. Please see the response to the AhNq and DSTd as well.
>
> [11] Fujimoto, Scott, Herke Hoof, and David Meger. "Addressing function approximation error in actor-critic methods." International conference on machine learning. PMLR, 2018.
>
>
> ### **Question 3. The benefit of Fourier feature network**
> **Intuitive explanation**: This is because quantile regression requires "the whole information" of the random variable, not a point information (e.g., median or expectation) of the random variable.
>
> **Theoretical explanation**: In quantile regression, the pointwise convergence in frequency space is required; i.e., high-frequency affects a lot. The requirement of "whole information" implies that the learned random variable converges in the distribution sense, and the Levy convergence theorem [12] tells that it is equivalent to the Fourier transform of the learned random variable converges pointwise.
>
>
> First, please note that learning the quantile function of a random variable $Z$ is equivalent to learning the Fourier transform of a random variable, $\varphi_{Z}$ (i.e., Characteristic function) by the following equation
> $$ \varphi\_{Z}(t) = \mathbb{E}[\exp[itX]] = \int\_{\mathbb{R}} \exp(itx) dF_{Z}(x) = \int\_{0}^{1} \exp(it F^{-1}\_{Z}(p)
> )dp$$
>
> where $i = \sqrt{-1}$. Let $Z_{n} = F^{-1}_{Z_n}(U), U \sim [0, 1]$ be a random variable corresponding learned random variable corresponding the learned quantile function at learning step $n$.
>
> For the risk-sensitive RL, the $n$-step learned quantile $Z_{n}$ is required to converge in distribution to $Z$, contrary to expectation regression (MSE) or median regression (MAE). Levy convergence theorem [12] tells that $\varphi_{Z_n}$ converges pointwise to $\varphi\_{Z}$ if and only if $Z_{n}$ converge in distribution to $Z$; i.e., high frequency affects the convergence.
>
> [12] Williams, D. (1991). Probability with Martingales. Cambridge University Press. section 18.1. ISBN 0-521-40605-6.
>
> ### **Question 4. Ensemble-size Ablation**
> We added the experiment results in Appendix A. As shown in TQC [13], performance saturation was observed as we increased the number of critics by nearly 5-7. Since the experimental result does not provide any new insights compared to the previous argument in TQC, we decided to report the result in the Appendix due to the margin issue. However, we are willing to include the result in the main text if the reviewer requires it.
>
> [13] Kuznetsov, Arsenii, et al. "Controlling overestimation bias with truncated mixture of continuous distributional quantile critics." International Conference on Machine Learning. PMLR, 2020.

---

> ### Comment · Reviewer_3isb · 2024-11-22
>
> Thanks for your rebuttal. As you have mentioned, compared with Bernard et al.'s work, this work carefully selects/estimates $\mu$, $\sigma$, and $\varepsilon$ while Bernard et al.'s work requires the ground truth values. It is reasonable as these ground truth values are hard to calculate in RL. So are there any results showing that these estimations are unbiased? Or if there is an estimation error, what is the gap between the estimated risk and the ground truth risk? I think deeper discussions (theoretically or experimentally) about these questions will help this paper become more solid.

---

> ### Author Response · Authors · 2024-11-22
> **Response to the further question**
>
> Thank you for your response.
> Since there is no standard criterion for the tolerance ($\varepsilon$), we will talk about mean ($\mu$) and standard deviation ($\sigma$).
>
> ### **Question. Are these estimations unbiased?**
> Yes, according to Koenker [1, 2], the quantile regression will yield an unbiased estimator (in a discounted sense). Dabney [2]'s discussion tells us that convergence is guaranteed as well.
>
> We have updated the discussion about unbiasedness (revised version 370) and the effect of Fourier feature networks in a theoretical sense (revised versions 374-377).
>
> We hope we have addressed all of the reviewer's concerns.
>
>
> [1] Koenker, R., and Hallock, K. 2001. Quantile Regression: An Introduction. Journal of Economic Perspectives 15(4):4356.
>
> [2] Dabney, Will, et al. "Distributional reinforcement learning with quantile regression." Proceedings of the AAAI conference on artificial intelligence. Vol. 32. No. 1. 2018.
>
> **P.S** . The discussion of how discounting affects the overall results is an interesting topic for future research, but it is out of scope for this work.

---

> > ### Comment · Reviewer_3isb · 2024-11-24
> >
> > Thanks for your response. I keep positive about this work.

---

### Official Review · Reviewer_mn9s · 2024-10-31

**Soundness:** 2
**Presentation:** 2
**Contribution:** 2
**Rating:** 6
**Confidence:** 3

**Summary:**

This paper introduces MR-IQN, a model-risk-sensitive offline RL algorithm, which aims to minimize risk in worst-case scenarios instead of conventional risk. Building on TD3+BC, it incorporates a critic ensemble to estimate model risk and employs a Fourier feature quantile regression network to mitigate spectral bias, allowing for precise distributional statistics estimation. MR-IQN outperforms baseline algorithms CODAC and ORAAC in two finance environments, Forex and Stock, as well as in the Airsim environment.

**Strengths:**

1. The approach of minimizing model risk in offline RL using a critic ensemble, rather than focusing on conventional risk, is an interesting and novel idea.
2. MR-IQN demonstrates strong results in the presented environments, indicating promising potential.

**Weaknesses:**

1. The experimental section feels underdeveloped, as MR-IQN and the baselines are only tested in three environments. Expanding the evaluation to include a broader set of benchmarks, such as those in D4RL (as done in CODAC and ORAAC), would significantly strengthen the results.
2. The choice of metrics in Table 1 raises some concerns. Evaluating performance at the 50% quantile is unusual, as this metric doesn’t emphasize rare events, which seems at odds with the motivation of risk-sensitive RL. To the best of my knowledge, previous literature typically uses the 10% quantile. Could you clarify why CV@R(50%) was selected for this study?
3. The explanation of model risk is somewhat unclear, and Figure 1 is particularly challenging to interpret. A more detailed and structured presentation of model risk concepts would improve readability and understanding.

**Questions:**

See above

---

> ### Author Response · Authors · 2024-11-17
>
> We appreciate your valuable feedback.
>
> ### **Weakness 1. Lack of Experiment (D4RL)**
>  We had reported the D4RL results in Appendix A.1. Following the concern, we added a guide to find out the D4RL results in the main text (revised version 449-450 lines). Here are the summarized results. For the details, please see Appendix A.1. in the supplementary material.
>
> | Algorithms  | Average | Average without Random |
> |-----------|-----------|-----------|
> | $\text{MR-IQN}\_{\text{Wang} -0.5}$ | 68.59 | 85.44|
> |$\text{1R2R}_{\text{Wang}}$| 67.35 | 81.43 |
> |$\text{1R2R}_{\text{CV@R}}$| 67.78 | 81.99|
> |ORAAC| 31.83 | 36.08 |
> |CODAC| 61.34 | 75.90 |
> |TD3+BC| 58.86 | 76.13|
>
> Here ***Average*** means average over {Random, Medium, Medium Replay, Medium Expert} of {Halfcheetah, Hopper, Walker2d} environments.
>
> and   ***Average without Random*** means average over  {Medium, Medium Replay, Medium Expert} of {Halfcheetah, Hopper, Walker2d} environments.
>
> ### **Weakness 2. The reason why CV@R 10% is not chosen for the evaluation metric**
>
>  As the reviewer requested, we report the CV@R 10\% score here.
>
> |          |               | Target            | Risk Measures  |           |               | Target | Risk Measures |
> |--------|-------------|-----------|-----------|--------|-------------|-----------|-----------|
> |Env   | Algorithm| CV@R (50%)| Wang (-0.5)       | Env   | Algorithm| CV@R (50%)| Wang (-0.5)|
> |  |CODAC  | $-48.72 \pm 25.25$ | $-51.27 \pm 22.79$ |   |CODAC  | $2.77 \pm 1.51$ | $3.00 \pm 0.91$ |
> | Forex |ORAAC| $-100.0 \pm 0.0$ | $-96.59 \pm 1.04$ |  Stock |ORAAC| $7.85 \pm 0.82$ | $7.92 \pm 0.86$  |
> | (High) |IQN TD3+BC | $-11.47 \pm 28.97$ | $-27.91 \pm 48.02$| (Low) |IQN TD3+BC | $8.33 \pm 0.67$ | $8.32 \pm 0.71$ |
> |  | MR-IQN (ours) | $25.71 \pm 32.09$| $9.44 \pm 59.11$| | MR-IQN (ours) | $10.06 \pm 0.53$ | $10.14 \pm 0.55$ |
>
> The tendency of CV@R 10\% did not significantly differ from other evaluation metrics; thus, it did not provide additional insights.
> This is attributed to the highly dynamic nature of the environment, where a momentary wrong decision can significantly affect the total return of an episode. These momentary choices determine the overall episode return, leading to high variability in total returns. Consequently, rare but adverse events essentially determine other evaluation metrics like the mean and CV@R(50\%).
>
> We reported the target risk measure because we wanted to demonstrate that the learning model risk makes the policy robust with respect to the learned risk measure (target risk measure).
> However, we are willing to report a CV@R 10\% score if the reviewer requests.
>
> ### **Weakness 3. Revise Figure 1.**
>  We revised Figure 1 to explain the concept. Please see Page 2. in the revised version.

---

> > ### Comment · Reviewer_mn9s · 2024-11-22
> >
> > Thank you for providing the additional experiments and scores.
> >
> > I’d like to clarify the interpretation of the reported CV@R 10% scores. While the reported scores are CV@R 10%, the target risk measures for optimization appear to be CV@R 50% and Wang(-0.5), which differ from the evaluation metrics. Is this understanding correct? If so, a natural follow-up question is whether MR-IQN’s performance is sensitive to the choice of the target risk measure. For example, in my opinion, including experiments where CV@R 10% is used as both the target risk measure and evaluation metric would strengthen the paper.

---

> ### Author Response · Authors · 2024-11-24
> **Response for further question and suggestion.**
>
> We appreciate your response. The experimental results have finally been obtained.
> Now, we are analyzing the results and will add the experiment results with detailed analysis to the Appendix as soon as possible.
>
> ### **Question 1: Are the reported scores CV@R 10%, while the optimization uses different target risk measures that differ from the evaluation metrics?**
> Yes, the previously reported results are based on the target risk measure mentioned in the main text.
>
> ### **Question 2: Is MR-IQN's performance sensitive to the choice of target risk measure?**
> Yes, not only MR-IQN but also any risk-sensitive RL performances are sensitive to the choice of target risk measure.
> Please note that the objective of MR-IQN is
> $$\text{minimize } \text{MR}\_{\phi}(Z^{\pi}(s,a);\mu, \sigma, \varepsilon) \text{such that } D(\pi || \pi\_{\mathcal{D}}) < \epsilon,$$
> as presented in Eq. (10) of the main text.
>
> Meanwhile,
> $$\text{MR}\_{\phi}(Z^{\pi}(s,a);\mu, \sigma, \varepsilon) =  \\max\_{X \in M(Z^{\pi}(s,a); \mu, \sigma, \varepsilon)} H\_{\phi}(X) =  \\max\_{X \in M(Z^{\pi}(s,a); \mu, \sigma, \varepsilon)} \int\_{0}^{1} F^{-1}\_{X}(p) \phi(p) dp.$$
> Here, informally, $M(Z^{\pi}; \mu, \sigma, \varepsilon)$  is a set of random variables that are close enough to  $Z^{\pi}(s,a)$ and $X$ is the close enough random variable.
> Since we finally measure the risk with $H\_{\phi}(X) = \int_{0}^{1} F^{-1}\_{X}(p) \phi(p) dp$, which is $\phi$ (the target risk measure) dependent.
>
> On the other hand, traditional risk-sensitive RL minimizes the risk defined as
> $$H\_{\phi}(Z^{\pi}(s,a)) = \int_{0}^{1} F^{-1}\_{Z^{\pi}(s,a)}(p) \phi(p) dp$$
> which is also $\phi$ dependent.
>
>
> ### **Suggestion: Including experiments where CV@R 10% is used as both the target risk measure and evaluation metric would strengthen the paper.**
> We are analyzing the results and will first present our results.
>
> |                       | Target Risk Measure   | CODAC            | ORAAC            | IQN-TD3+BC       | MR-IQN (ours)    |
> |-----------------------|-----------------------|------------------|------------------|------------------|------------------|
> |    **Stock Env.**    | *Wang*(-0.5)         | $3.00 \pm 0.91$  | $7.92 \pm 0.86$  | $8.32 \pm 0.71$  | $10.14 \pm 0.55$ |
> |                      | *CV@R*(50%)          | $2.71 \pm 1.51$  | $7.85 \pm 0.82$  | $8.33 \pm 0.67$  | $10.06 \pm 0.53$ |
> |                       | *CV@R*(10%)          | $3.98 \pm 0.86$  | $7.91 \pm 0.74$  | $8.68 \pm 0.43$  | $10.35 \pm 0.52$ |
> | **Forex Env.**        |  *Wang*(-0.5)         | $-51.27 \pm 22.79$ | $-96.59 \pm 1.04$ | $-27.91 \pm 48.02$ | $9.44 \pm 59.11$ |
> |                       |  *CV@R*(50%)          | $-48.72 \pm 25.25$ | $-100 \pm 0.00$ | $-11.47 \pm 28.97$ | $25.71 \pm 32.09$ |
> |                       | *CV@R*(10%)          | $-4.35 \pm 2.06$  | $-47.54 \pm 20.94$ | $0.25 \pm 34.37$ | $31.40 \pm 36.02$ |
>
> We are working day and night to strengthen the paper and will provide the analysis along with the revised version as soon as possible.
>
> **P.S.** We slightly abused the $\max$ operator instead of $\sup$ for better understanding.

---

> > ### Comment · Reviewer_mn9s · 2024-11-25
> >
> > Thank you for providing these additional experiments. Overall, I find the idea behind MR-IQN to be both interesting and a valuable contribution to the community. I look forward to reviewing your revision, and I am inclined to recommend acceptance, provided the additional experiments and analysis are satisfactory.

---

> ### Author Response · Authors · 2024-11-25
> **Suggestion: Including experiments where CV@R 10%**
>
> Finally, we have completed the analysis of the experiment results. Due to the margin issue, we added this analysis in Appendix A.4. in the revised version. Instead, we added a comment in the main text to indicate the presence of this analysis (revised version, line 450).
>
> ### **Analysis of CV@R(10%) Results**.
>
> |                       | Target Risk Measure   | CODAC            | ORAAC            | IQN-TD3+BC       | MR-IQN (ours)    |
> |-----------------------|-----------------------|------------------|------------------|------------------|------------------|
> |    **Stock Env.**    | *Wang*(-0.5)         | $3.00 \pm 0.91$  | $7.92 \pm 0.86$  | $8.32 \pm 0.71$  | $10.14 \pm 0.55$ |
> |                      | *CV@R*(50%)          | $2.71 \pm 1.51$  | $7.85 \pm 0.82$  | $8.33 \pm 0.67$  | $10.06 \pm 0.53$ |
> |                       | *CV@R*(10%)          | $3.98 \pm 0.86$  | $7.91 \pm 0.74$  | $8.68 \pm 0.43$  | $10.35 \pm 0.52$ |
> | **Forex Env.**        |  *Wang*(-0.5)         | $-51.27 \pm 22.79$ | $-96.59 \pm 1.04$ | $-27.91 \pm 48.02$ | $9.44 \pm 59.11$ |
> |                       |  *CV@R*(50%)          | $-48.72 \pm 25.25$ | $-100 \pm 0.00$ | $-11.47 \pm 28.97$ | $25.71 \pm 32.09$ |
> |                       | *CV@R*(10%)          | $-4.35 \pm 2.06$  | $-47.54 \pm 20.94$ | $0.25 \pm 34.37$ | $31.40 \pm 36.02$ |
>
> **Analysis**
>
> In Stock environment, only CODAC demonstrates a significant improvement in *CV@R*(10%) risk, whereas the performance of other algorithms appears to have reached saturation.
>
> In Forex environment, all algorithms exhibit consistent behavior, with *CV@R*(10%) optimization yielding the highest *CV@R*(10%) performance, followed by *CV@R*(50%) and *Wang*(-0.5) optimization criteria, respectively, except for ORRAC, which does not show meaningful *CV@R*(10%) performance in either *CV@R*(50%) or *Wang*(-0.5) risks.
>
> To examine this phenomenon, we calculated the similarity among risk measures using Wasserstein 2 distance ($D\_{W;2}$). The $D\_{W;2}$ between *CV@R*(50%) and  *CV@R*(10%) is approximately 0.23, while $D\_{W;2}$ between *Wang*(-0.5) and  *CV@R*(10%) is approximately 0.40. This indicates that *CV@R*(50%) is more *similar* risk measure with *CV@R*(10%) compared to *Wang*(-0.5). This similarity (Wasserstein distance) explains the phenomenon; Optimizing with respect to *CV@R*(50%) tends to yield superior performance of  *CV@R*(10%) compared to optimizing *Wang*(-0.5)
>
>
> **Why is Wasserstein distance appropriate to calculate the similarity?**
>
> Please note that spectral risk measure is defined as
> $$H\_{\phi} (Z) = -\int\_{0}^{1}F^{-1}_{Z}(p) \phi(p) dp.$$
> In the definition, $\phi$ governs the equation. Therefore, we can identify the risk measure with the weight function $\phi$, which belongs to a probability density function defined on [0, 1]. Since $\phi$ is a probability density function, we can calculate the Wasserstein distance between the risk measures with $\phi$.
>
> Meanwhile, Wasserstein distance provides a more natural interpretation of the similarity between risk measures compared to KL-divergence and extendible for other classes of risk measure (e.g., *Value at Risk, V@R*). V@R measure can be written in a similar form as
> $$\textit{V@R}(Z; \alpha) = -\int\_{0}^{1} F^{-1}_{Z}(p) \delta(p - \alpha) dp = -F^{-1}\_{Z}(p).$$
> The difference between $\textit{V@R}(Z; \alpha)$ and worst-case risk measure for very small $\alpha$  should be proportional to $\alpha$, but $f$-diveregence cannot estimate their similarity. While Wasserstein distance yields the desired results (the difference is proportional to $\alpha$).
>
> The revised version of Appendix A.4 has detailed explanations about it with visualizations of risk measures.
>
> **P.S.** The reason why we have introduced the *V@R*, which does not belong to spectral risk measure, is for more intuitive explanation. You may get similar results comparing the $f$-divergence between the *worst-case* risk measure and *Wang* risk measure, and both of them belong to the spectral risk measure.

---

> > ### Comment · Reviewer_mn9s · 2024-11-29
> >
> > Thank you for providing these new results and analysis. My concerns have been addressed, and I have accordingly raised the score.

---

> > > ### Author Response · Authors · 2024-11-29
> > >
> > > We thank you for your response and welcome any further questions until the final day we can respond.

---

### Official Review · Reviewer_AhNq · 2024-11-02

**Soundness:** 2
**Presentation:** 3
**Contribution:** 2
**Rating:** 6
**Confidence:** 3

**Summary:**

This work introduces a framework and learning algorithm, MR-IQN, for offline reinforcement learning in a risk-sensitive context. The MR-IQN algorithm aims to identify a policy that minimizes model risk while remaining close to the behavior policy derived from the dataset. The authors evaluate the algorithm on both finance and self-driving datasets, demonstrating that it outperforms baseline methods in terms of model risk measures and average return.

**Strengths:**

1. **Novelty:** To the best of my knowledge, this is a novel formulation for risk-sensitive offline RL, with no similar approach appearing in current literature. Additionally, the use of a critic ensemble and a Fourier feature network for variance and bias reduction is a thoughtful design choice.
2. **Clarity:** The paper is well-organized and self-contained, providing a clear presentation of the problem, algorithmic approach, and experimental setup.
3. **Comprehensive Experiments:** The evaluation on finance and self-driving applications aligns well with the risk-sensitive focus of the paper. Extensive experiments, including a range of baselines, lend further credibility to the results.

**Weaknesses:**

**Major Issues:**

1. The formulation of risk-sensitive offline RL optimizes model risk regularized by the dataset's behavior policy. It is unclear how this formulation guarantees that the optimality considers both model risk and expected return, especially when the behavior policy in the dataset may not be optimal, as is often the case in offline RL.
2. There is a lack of theoretical insight into the algorithm. Specifically, it is not evident why the proposed loss function in Equation (11) adequately solves the problem defined in Equation (10). Additional theoretical justification would enhance understanding of the algorithm's effectiveness.

**Suggested Improvements:**

1. In the experimental section, an ablation study on ensemble size (i.e., number of critics) would help substantiate the value of using an ensemble for performance gains.
2. The authors choose $\mu$ as the minimum value among critics' estimations and $\sigma$ as the maximum, aiming for conservatism. It would strengthen the argument if they defined conservatism and justified the choices of $\mu$ and $\sigma$.

**Minor Issues:**

The following suggestions may improve clarity:

1. In Equation (3), the form $ F^{-1}_{\phi}(U[0, 1]) $ is not clearly defined by Definition 1.
2. Is $ Z_{\pi} $ intended to mean the same as $ Z^{\pi} $? If so, consistent notation is recommended.
3. In lines 275-276, the notation $ E_{p_j} $ appears undefined or unclear.
4. In Figure 4, the x-axis and y-axis require explicit labeling for clarity.
5. In Tables 1 and 2, the meaning of "Average" should be clarified.

**Questions:**

1. I appreciate the illustrative explanation of the calculation for $ F_{ens.}^{-1} $; however, is there a mathematical formula for this calculation? Additionally, were gradients preserved when calculating $ F_{ens.}^{-1} $ during experiments?
2. In the experiments, how was the offline dataset generated?

---

> ### Author Response · Authors · 2024-11-17
> **Weakness**
>
> We appreciate your constructive feedback.
> ### **Weakness. 1.1 It is unclear how minimizing model risk guarantees both model risk and expected return**
> Our proposed approach does not focus on optimal expected return but on minimizing the risk. As shown in the Stock environment case in Table 1, there was a degraded expected return compared to other baselines. However, there is a close relationship between minimizing risk and maximizing expected return considering the distribution shift.
>
> ***Detailed explanation about model risk minimization***
>
> Minimizing model risk is important because our policy must make robust decisions under incomplete models (i.e., critics), which have a significant possibility of errors stemming from various reasons, such as the "deadly triad"[1], distribution shift[2], correlation error[3], and so on. Making robust decisions based on this incomplete estimation is the best possible approach, and model risk provides a solution to ensure robustness under this incompleteness [4].
>
> ***Intuitive explanation about the relationship between risk minimization and expected return maximization***
>
> When the environment is extremely dynamic (e.g., the Forex case), the probability of the model being wrong becomes higher, and the wrong critics will lead to degraded performance in both expected return and risk.
> Meanwhile, our approaches admit the possibility that the critics are wrong, and the actor is trained to deal with the worst-case situation when the critics are wrong. As a result, our approach remains robust in expected return in highly dynamic environment cases.
>
> ***Theoretical explanation***
>
> Theoretically, risk minimization is maximizing the worst-case of expected return among plausible alternative scenarios; that is, minimizing the risk is maximizing the lower bound of the expected return. Therefore, we can expect the expected return to have an increasing tendency when we minimize the risk. Further, the model risk-sensitive offline RL chooses the lower bound among broader alternative scenarios compared to traditional risk-sensitive offline RL. This explains why our approach shows better performance in expected returns in highly dynamic environments (e.g., Forex).
>
>  According to the analysis below, we can expect that the expected return of a model risk-sensitive approach will outperform traditional risk-sensitive offline RL whenever a distribution shift occurs, but its support is contained within the critic's estimation, and vice versa.
>
> Every spectral risk  can be also written as
> $$    H_{\phi}(Z^{\pi}(s, a)) =-\int\_{0}^{1} F^{-1}\_{Z^{\pi}}(p; s,a) \phi(p)dp =  \max_{\mathbb{Q}  \in \mathcal{T}(Z^{\pi};\phi)} (-\mathbb{E}_{\mathbb{Q}}[Z^{\pi}(s, a)]), $$
>
> where $\mathcal{T}(Z^{\pi};\phi)$ is the plausibility constraint determined by risk measure $\phi$, and $\mathcal{T(Z^{\pi};\phi)}$ is subsets of probability measures whose support are contained in the support of $Z^{\pi}(s,a)$ [5] .
> The direct interpretation of above equation is the minimal value of "expected return" when the distribution shift whose occurs, while the critics' support is still remaining valid.
>
> Meanwhile, our objective, model risk allows different support and finds the worst case in a broader space,
> $${MR}_{\phi}(Z^{\pi}(s, a); \mu, \sigma, \varepsilon)=\\max\_{X \in M(Z^{\pi}; \mu, \sigma, \varepsilon)} H\_{\phi}(X) = \\max\_{X \in M(Z^{\pi}; \mu, \sigma, \varepsilon)} \max\_{\mathbb{Q}  \in \mathcal{T}(Z^{\pi};\phi)} (-\mathbb{E}\_{\mathbb{Q}}[Z^{\pi}(s, a)]) = \max\_{\mathbb{Q} \in \mathcal{Q}}(-\mathbb{E}\_{\mathbb{Q}}[X(s, a)])  $$
> where $\mathcal{Q} = \bigcup\_{X \in M(Z^{\pi}; \mu, \sigma, \varepsilon)} \mathcal{T}(X;\phi)$. The last representation is just for simplifying two $\max$ operators to the single $\max$ operator.
>
> These expectation representations above allow us to gain insight into how minimizing risk helps to increase expected returns.
> However, please recall that these equations concern the "worst-case" scenario, not the real distribution shift; therefore, we cannot strictly guarantee expectation maximization.
>
> ***P.S***. We slightly abused the $\max$ operator instead of $\sup$ for better understanding.
>
>  [1] Van Hasselt, Hado, et al. "Deep reinforcement learning and the deadly triad." arXiv preprint arXiv:1812.02648 (2018).
>
> [2] Fu, Justin, et al. "Diagnosing bottlenecks in deep q-learning algorithms." International Conference on Machine Learning. PMLR, 2019.
>
> [3] Fujimoto, Scott, et al. "Why Should I Trust You, Bellman? The Bellman Error is a Poor Replacement for Value Error." International Conference on Machine Learning. PMLR, 2022.
>
> [4] Breuer, Thomas, and Imre Csiszár. "Measuring distribution model risk." Mathematical Finance 26.2 (2016): 395-411.
>
> [5] Föllmer, Hans; Schied, Alexander (2004). Stochastic finance: an introduction in discrete time (2 ed.). Walter de Gruyter. ISBN 978-3-11-018346-7.

---

> ### Author Response · Authors · 2024-11-17
> **Weakness Cont.**
>
> ### **Weakness 1.2. Policy Improvement**
> Please recall the policy loss function is
> $$  \mathcal{L}(\pi) = \lambda\_{\text{q. learn}} \text{MR}(Z^{\pi}; s, a) +  (\pi(s) - a\_{\mathcal{D}})^2. $$
> The intuitive explanation is that given dataset action $a_{\mathcal{D}}$, the actor will behavior clone the dataset action if the model risk is sufficiently small and won't generate the action if the model risk is high. As the behavior clone is enforced by MSE loss, the slight modification of action is also allowed when the benefit of reducing model risk is higher. This design is directly followed by the traditional offline RL algorithm, TD3+BC [6].
>
> ### **Weakness 2. How does Eq. 11 align with Eq. 10?**
>
> This is an auxiliary objective proposed by Fujimoto et al. [6].
> Please note that if the data-set action is unimodal, the MSE will reduce the KL divergence between the actor's action, achieving as follows:
> $$
>     \pi^{*} = \text{arg}\min \lambda\_{\text{q.learn}} \text{MR}\_{\phi}(Z^{\pi}(s,a); \mu, \sigma, \varepsilon) + (a - a\_{\mathcal{D}})^{2}  =
>     \text{arg}\min \lambda\_{\text{q.learn}} \text{MR}\_{\phi}(Z^{\pi}(s,a); \mu, \sigma, \varepsilon) + D\_{KL}(a \lVert  a\_{\mathcal{D}})
> $$
> $$     = \text{arg}\min  \text{MR}\_{\phi}(Z^{\pi}(s,a); \mu, \sigma, \varepsilon) \text{ such that} D\_{KL}(a \lVert  a\_{\mathcal{D}}) < \epsilon({\lambda\_{\text{q.learn}}}). $$
>
> This approach is quite common in offline RL, including decision transformer [7], BCQ [8], and diffusion QL [9]. The paper of diffusion QL [4] provides well-organized details. Also, please notice that the unimodal distribution of the dataset doesn't have to be enforced for all actions. It allows a slight degree of multimodality.
>
>
> [6]  Fujimoto, Scott, and Shixiang Shane Gu. "A minimalist approach to offline reinforcement learning." Advances in neural information processing systems 34 (2021): 20132-20145.
>
> [7] Chen, Lili, et al. "Decision transformer: Reinforcement learning via sequence modeling." Advances in neural information processing systems 34 (2021): 15084-15097.
>
> [8] Fujimoto, Scott, David Meger, and Doina Precup. "Off-policy deep reinforcement learning without exploration." International conference on machine learning. PMLR, 2019.
>
> [9] Wang, Zhendong, Jonathan J. Hunt, and Mingyuan Zhou. "Diffusion Policies as an Expressive Policy Class for Offline Reinforcement Learning." The Eleventh International Conference on Learning Representations.
>
> ### **Suggested Improvements 1. Ensemble-size ablation study**
> We just updated Appendix. A. regarding the requested experiment.
> As shown in TQC [10], performance saturation was observed as we increased the number of critics by nearly 5-7.
> Since the experimental result does not provide any new insights compared to the previous argument in TQC, we decided to report the result in the Appendix due to the margin issue.
> However, we are willing to include the result in the main text if the reviewer requires it.
>
> [10] Kuznetsov, Arsenii, et al. "Controlling overestimation bias with truncated mixture of continuous distributional quantile critics." International Conference on Machine Learning. PMLR, 2020.
>
> ### **Suggested Improvements 2. Strengthened argument about choice of $\mu$ and $\sigma$**
>
> There are two main reasons to choose minimal $\mu$ and maximal $\sigma$.
> * To minimize overestimation bias in actor-critic methods, following the findings of Fujimoto et al. [11]
> *  To reduce the edge case, when $\varepsilon < (\mu\_{Z} - \mu)^{2} + (\sigma\_{Z} - \sigma)^{2}$ (equation in Theorem 1.) (i.e., when the model risk does not have a solution).
>
> ***Reducing overestimation bias:*** Fujimoto et al. [11]. have verified that using minimal expected value ($\mu$) reduces the overestimation bias in online RL. The maximal standard deviation ($\sigma$ ) is a direct analogy of choosing minimal $\mu$, as large standard deviation $\sigma$ is something harmful.
>
> ***Reducing Edge Case***: As we choose the minimal $\mu$, and maximal $\sigma$ for calculating model risk, the edge case occurs
> when one of the $\mu$ or $\sigma$ is not chosen from the reference distribution, which is the most conservative critic as
>
> $$F^{-1}\_{\text{ref}}(\cdot) = F^{-1}\_{Z^{\pi}}(\cdot; s, a, \theta^{(\*)}), \text{ where } \theta^{(\*)} = \text{argmax}\_{\theta^{(i)}} H\_{\phi}(Z^{\pi}(s, a, \theta^{(i)})).$$
>
> However, since the reference distribution is the most conservative one, except in the very early stage of the training, the probability both $\mu$ and $\sigma$ are chosen from the reference distribution is high, making $(\mu_{Z} - \mu)^{2} + (\sigma_{Z} - \sigma)^{2} = 0$ and prevent the edge case. This is because high risk implies a low mean and high standard deviation in general.
>
> We added the citation of [11] in the revised version line 292-293.
>
> [11] Fujimoto, Scott, Herke Hoof, and David Meger. "Addressing function approximation error in actor-critic methods." International conference on machine learning. PMLR, 2018.

---

> ### Author Response · Authors · 2024-11-17
> **Questions and Minors**
>
> ### **Question 1.1 Math Formula for $F^{-1}\_{\text{ens.}}$**
> Mathematically, the explicit formula is known only for the very special case (and does not exist in some cases) [12]. Please remind that the existence of a formula does not determine the existence of a function. Although there might not be a formula, the function always exists.
>
> We introduce the formula when the special case; there are only two critics. Let $X$ be a random variable whose quantile is $F^{-1}\_{\text{ens}}$. We can formulate the $X = B Z^{\pi}_{1} + (1 - B) Z^{\pi}\_{2}$ where $B$ is the Bernoulli distributed random variable with parameter $0.5$ by inverse sampling transform.
>
> Define
> $\alpha\_{\*} := \inf \\{\alpha \in (0, 1) | \exists \beta \in (0,1)  \text{ such that } \frac{1}{2}(\alpha + \beta) = p \\ \text{ and } F^{-1}\_{Z^\pi}(\alpha; \theta^{(1)}) \ge  F^{-1}_{Z^\pi}(\beta; \theta^{(2)}) \\}$
>
> when the argument set is non-empty. Define $\beta\_{\*}=2(p - \frac{1}{2} \alpha\_{\*}) \in [0,1]$ (i.e., the $\beta$ corresponding $\alpha_{\*}$). The formula is
>
>
> $$ F^{-1}\_{\text{ens}}(p) = \max\\{F^{-1}\_{Z^\pi}(\alpha\_{\*}; \theta^{(1)}), F^{-1}\_{Z^\pi}(\beta\_{\*}; \theta^{(2)}) \\} $$
>
> The explicit formula is impractical to calculate even for this simple case. For the detail please see [12].
>
> [12] Bernard, Carole, and Steven Vanduffel. "Quantile of a mixture." arXiv preprint arXiv:1411.4824 (2014).
>
>
>
> ### **Question 1.2 Math Formula for Gradient preservation of $F^{-1}\_{\text{ens.}}$**
>
> Creating $F^{-1}\_{\text{ens}}$ itself preserves the gradient, but we used a stop-gradient for $\varepsilon$, which is the only variable related to $F^{-1}\_{\text{ens}}$. Semantically, the error tolerance ($\varepsilon$) should not be minimized by the decision-maker but should be given by the external environment. That is why we used the stop-grad operator on $\varepsilon$.
>
> Additionally, please note that the JAX gradient is preserved through sorting, interpolation, and numerical integration operations, while PyTorch might require more careful implementation [13].
>
> [13] https://discuss.pytorch.org/t/differentiable-sorting-and-indices/89304
>
>
> ### **Question 2. How was the offline data generated?**
>
>  We generated the dataset using a trained but premature online RL algorithm (i.e., *medium* in D4RL crietrion). More details about the dataset information are provided in Appendix A.2.
>
> ### **Minors**
> 1. We updated the typo and added explanation about $F^{-1}_{\phi}(U)$ in the Definition 3. in revised version
> 2. $F^{-1}_{Z\pi}$ was intended because its font becomes too small to read when there is superscript in subscript, but we accepted the feedback and edited.
> 3. We also edited the algorithm's first and second lines (original paper lines 275-276) and added an explanation
> 4. We added an axis label in Figure 4.
> 5. We changed the label from Average to Mean to avoid confusion and added an explanation in Section 5.2. (revised version line 447).

---

> > ### Comment · Reviewer_AhNq · 2024-11-22
> >
> > Thanks for the rebuttal from the author. The response from the author has addressed my questions and concerns and I will update my rating.

---

> ### Author Response · Authors · 2024-11-24
>
> We appreciate your kind response, and additional questions are always welcome.

---

### Official Review · Reviewer_DSTd · 2024-11-03

**Soundness:** 4
**Presentation:** 3
**Contribution:** 3
**Rating:** 6
**Confidence:** 3

**Summary:**

This paper addresses the problem of model risk in offline reinforcement learning, which arises from distributional shifts in the learning dataset. The authors provide a critic ensemble framework, integrating well-researched predecessor methods into a critic ensemble for risk estimation that can be applied regardless of the transition distribution or differences in distributional support. Experimental results in two domains demonstrate the superiority of the method and highlight the tradeoffs between robustness and high performance.

**Strengths:**

The motivation is clear; the problem statement, issues with existing works, and connections to the provided solution are clean and easy to follow. I also appreciate the label brackets underneath parts of the equations.

The contribution addresses the issue both distributional shifts in data and model inaccuracies, which are critical challenges in offline RL.

**Weaknesses:**

Most of the formalizations are cited directly from previous works, with the exception of 4.1. Given that 4.1 is only half of the proposed methods section, this makes the contribution seem more incremental. Are there changes or improvements that can be made to any of these components to better suit them to the domain? As an example, the Bellamn equation in Eq. 4 could be extended to measures such as minimum reward, expected value (over actions), or regret.

There are a few minor spelling errors, like 'qunatile' in the Fig. 4 caption.

**Questions:**

1) It is mentioned that when $\epsilon \leq (\mu_z-\mu)^2 + (\sigma_z - \sigma)^2$ holds, the quantile function doesn't have a solution. Under what distribution conditions does this occur, and how often is it observed? It would be useful to know if the strictly-less-than case occurs with relative infrequency, or very often.

2) Baselines ORAAC, CODAC, and 1R2R are mentioned in the introduction and related works, but only the first two are used as comparison baselines. Is there a reason the most recent method was excluded?

---

> ### Author Response · Authors · 2024-11-17
>
> We appreciate your valuable feedback.
>
> ### **Weakness.1. What is the major challenge in applying Bernard et al.'s work to your methods?**
>
> What we have done is
>
> (a). Calculating proper gradients for the policy with model risk with inaccurate, but at best statistics $\mu$, $\sigma$.
>
> (b). Properly choose tolerance $\varepsilon$ so that it aligns well with its semantic and rarely makes the edge case (i.e., $\varepsilon < (\mu - \mu_{Z})^{2} + (\sigma - \sigma_{Z})$.)
>
> (c). Reducing the error of estimated statistics ($\mu$, $\sigma$, and risk) with Fourier feature network.
>
> ***Detailed explanations.***
>
> (a). Although Theorem 1 and Corollary 1 are directly borrowed from **financial mathematics** (Bernard et al.), they require unrealistic assumptions to directly apply the results to the RL domain.
> They require the ground truth mean ($\mu$), deviation ($\sigma$), and a well-tailored tolerance ($\varepsilon$), and it is usually impossible to calculate the exact ground truth mean and deviation in offline RL setting because of "deadly triad" [1]  and "distribution shift" [2],  both of which are almost inevitable in deep offline RL. Our framework provides the guidelines for choosing these statistics so that the policy gradient calculated by the model risk can align well with the RL objective.
>
> (b). Furthermore, model risk is not designed for machine learning; therefore, it might fail to remain valid in edge cases during training. Our framework is designed to select $\mu$, $\sigma$, and $\varepsilon$ to avoid edge cases as much as possible and includes stop-gradient tricks and edge-case handling methods. This enables us to provide proper signals during training, as the results have shown.
>
> (c). Meanwhile, in Fourier feature networks, there has been no study about their effects on quantile regression.
> As shown in Figure 4, the median could be estimated without the Fourier feature network (i.e., a conventional $ L^1$ regression problem). However, the errors across the entire quantile range were large, showing spectral bias in quantile regression. We emphasize these effects.
> We have argued that the Fourier feature network is essential rather than optional in a risk-sensitive RL setting with distributional RL: The study about using the Fourier feature network to RL has demonstrated that the Fourier feature is a control mechanism between stability-performance control mechanisms based on frequency space. However, in distributional RL, there are always "noise factors" as we cannot fully reduce the loss exactly $0$ because of the design of quantile regression loss. For the theoretical explanation, please see the response to 3isb Q. 3.
>
> ### **Weakness.2. Typo in Figure 4.**
>
> We appreciate it and fixed the typo in the revised version.
>
> ### **Question 1. When the edge case (i.e., the model risk does not exist: $\varepsilon < (\mu_{Z} - \mu)^{2} + (\sigma_{Z} - \sigma)^{2}$?**
>
> The condition is very hard to meet except during the very early training stage (less than $0.1\\%$ in our setting). For the condition to be met:
> 1. **Similarity Among Critics:** Most of the critics must show similar distributions.
> 2. **Presence of an Outlying Critic:** There must be at least one outlying critic.
> 3. **Selection of the Outlier for $\mu$ or $\sigma$:** The outlying critic's distribution must be chosen for determining $\mu$ or $\sigma$.
> 4. **Exclusion of the Outlier from the Reference Distribution:** When both the outlying critic's $\mu$ and $\sigma$ are chosen, the outlying critic must not be chosen for the reference distribution.
>
> Except during the very early stage, condition 2 is hard to meet because all critics are trained using the same TD targets and the same loss function. Even if conditions 2 and 3 are met, in most cases, the outlying critic shows the smallest mean or largest deviation. Then, it is often the pessimistic critic; therefore, it is chosen to be the reference distribution (condition 4 is not met). However, since the probability is not zero, we designed the algorithm to handle it when it happens.
>
> ### **Question 2. The reason why 1R2R does not belong to the baselines of the main experiments.**
>
> The main reason is the limitation of model-based algorithms. 1R2R, as a model-based algorithm, requires a known terminal function (for details, please see these implementations [3, 4]). However, the terminal function in the environments in the main text is determined by their own simulation engine (e.g., collision detector). That's why we could not provide the 1R2R results in the main results. Instead, in Appendix A.1, we provided the results compared with 1R2R in the D4RL environment.
>
> [1] Van Hasselt, Hado, et al. "Deep reinforcement learning and the deadly triad." arXiv preprint arXiv:1812.02648 (2018).
>
> [2] Fu, Justin, et al. "Diagnosing bottlenecks in deep q-learning algorithms." International Conference on Machine Learning. PMLR, 2019.
>
> [3] https://github.com/marc-rigter/1R2R
>
> [4] https://github.com/junming-yang/mopo

---

> ### Author Response · Authors · 2024-11-24
>
> We hope we have addressed all of your concerns, and additional questions are always welcome.

---

> > ### Comment · Reviewer_DSTd · 2024-11-25
> >
> > Thank you for your thorough explanations, I will maintain my accepting recommendation.

---

### Author Response · Authors · 2024-11-30
**General Response to All Reviewers**

We appreciate the valuable and constructive feedback provided by all reviewers-Reviewer DSTd, Reviewer AhNq, Reviewer mn9s, and Reviewer 3isb. We have completed the revisions to our paper based on the reviewers' feedback and our own efforts, and the final version has been uploaded. We hope the reviewers will carefully evaluate these updates and find that they address any remaining concerns. A detailed list of revisions is provided below:

---

**Main Manuscript Revisions**
* **Texts**
    * We have improved the related work section (3isb).
    * We corrected the typo in Eq. (3) and added an explanation using label braces (AhNq).
    * We elaborated on the mathematical explanations for the necessity of the Fourier feature network (3isb, DSTd).
    * We provided justification for the implementation, specifically explaining why we chose $\min \mu$ and $\max \sigma$ (AhNq, 3isb).
    * We added text to notify readers about the additional experiments included in the Appendix (DSTd, mn9s).
    * We slightly paraphrased and corrected typos to ensure the content fits within the margin.
* **Figure, Algorithm, and Table**
    * We revised Figure 1, adding a more intuitive explanation of model risk (mn9s).
    * We fixed the typo and added axis labels in Figure 4 (AhNq, DSTd).
    * We changed the table label from "Average" to "Mean" to avoid confusion (AhNq).
    * We added an explanation for the first line of the Algorithm (AhNq).

---

**Appendix**

* We added experimental results from an ablation study addressing the correlation between performance and ensemble size (3isb, AhNq).
* We included experimental results for *CV@R*(10%) along with a detailed analysis (mn9s).

---
We acknowledge that the discussion period is ongoing and welcome any additional feedback, questions, or responses regarding our submission. We will do our best to respond promptly and thoroughly. Once again, we sincerely thank all reviewers for their time and dedication.

---

### Meta-Review · Area_Chair_8N5P · 2024-12-20

**Metareview:**

The reviewers acknowledged that the paper studies an important problem of risk-sensitive offline RL and proposes a novel methodology to minimize the worst-case risk instead of the expected risk. However, the reviewers also raised several concerns and questions in their initial reviews; in particular, one concern was about how the proposed methodology differs from the prior work. We want to thank the authors for their responses and active engagement during the discussion phase. The reviewers appreciated the responses and have an overall positive assessment of the paper, though the final ratings for all the reviewers remain borderline. The reviewers have provided detailed feedback, and we strongly encourage the authors to incorporate this feedback when preparing an updated version of the paper.

**Additional Comments On Reviewer Discussion:**

The reviewers raised several concerns and questions in their initial reviews; in particular, one concern was about how the proposed methodology differs from the prior work. After the discussion, all the reviewers have an overall positive assessment of the paper, though, the final ratings for all the reviewers remain borderline.

---

### Decision · Program_Chairs · 2025-01-22

Accept (Poster)